# ADAPTIVE WIDTH NEURAL NETWORKS

**Federico Errica**
NEC Laboratories Europe
federico.errica@neclab.eu

**Henrik Christiansen**
NEC Laboratories Europe

**Viktor Zaverkin**
NEC Laboratories Europe

**Mathias Niepert**
University of Stuttgart
NEC Laboratories Europe

**Francesco Alesiani**
NEC Laboratories Europe

## ABSTRACT

For almost 70 years, researchers have typically selected the width of neural networks' layers either manually or through automated hyperparameter tuning methods such as grid search and, more recently, neural architecture search. This paper challenges the status quo by introducing an easy-to-use technique to learn an *unbounded* width of a neural network's layer *during training*. The method jointly optimizes the width and the parameters of each layer via standard backpropagation. We apply the technique to a broad range of data domains such as tables, images, text, sequences, and graphs, showing how the width adapts to the task's difficulty. A by product of our width learning approach is the easy truncation of the trained network at virtually zero cost, achieving a smooth trade-off between performance and compute resources. Alternatively, one can dynamically compress the network until performances do not degrade. In light of recent foundation models trained on large datasets, requiring billions of parameters and where hyper-parameter tuning is unfeasible due to huge training costs, our approach introduces a viable alternative for width learning.

## 1 INTRODUCTION

Since the construction of the Mark I Perceptron machine (Rosenblatt, 1958) the effective training of neural networks has remained an open research problem of great academic and practical value. The Mark I solved image recognition tasks with a layer of 512 *fixed* "association units" that in modern language are the hidden units of a Multi-Layer Perceptron (MLP). MLPs possess universal approximation capabilities when assuming *arbitrary width* (Cybenko, 1989) and sigmoidal activations, and their convergence to good solutions was studied, for instance, in Rumelhart et al. (1986) where the backpropagation algorithm was introduced as "simple, easy to implement on parallel hardware", and improvable by other techniques such as momentum that preserve locality of weight updates.

Yet, after almost 70 years of progress (LeCun et al., 2015), the vast majority of neural networks, be they shallow or deep, still rely on a fixed choice of the number of neurons in their hidden layers. The width is typically treated as one of the many hyper-parameters that have to be carefully tuned whenever we approach a new task (Wolpert, 1996). The tuning process has many names, such as model selection, hyper-parameter tuning, and cross-validation, and it is associated with non-negligible costs: Different architectural configurations are trained until one that performs best on a validation set is selected (Mitchell, 1997). The configurations' space grows exponentially in the number of layers, so practitioners often resort to shortcuts such as picking a specific number of hidden units for *all* layers and trying a few values, which greatly reduces the search space together with the chances of selecting a better architecture for the task. Other techniques to tune hyper-parameters include constructive approaches (Fahlman & Lebiere, 1989; Wu et al., 2020), which alternate parameter optimization and creation of new neurons, natural gradient-based heuristics that dynamically modify the network (Mitchell et al., 2023), bi-level optimization (Franceschi et al., 2018), and neural architecture search (White et al., 2023), which often requires separate training runs for each configuration.

The hyper-parameters' space exploration problem is exacerbated by the steep increase in size of recent neural architectures for language (Brown et al., 2020) and vision (Zhai et al., 2022), for example, where parameters are in the order of billions to accommodate for a huge dataset. Training these models requires an amount of time, compute power, and energy that currently makes it unfeasible for most institutions to perform a thorough model selection and find good width parameters; the commonly accepted compromise is to stick to previously successful hyper-parameter choices. This may also explain why network pruning (Blalock et al., 2020; Mishra et al., 2021), distillation (Zhang et al., 2019) and quantization (Mishra et al., 2021) techniques have recently been in the spotlight, as they trade-off hardware requirements and performance.

This work introduces a *simple* and *easy to use* technique to learn the width of each neural network's layer without imposing upper bounds (we refer to it as **unbounded width**). The width of each layer is *dynamically* adjusted during backpropagation (Paszke et al., 2017), and it only requires a slight modification to the neural activations that does not alter the ability to parallelize computation. The technical strategy is to impose a soft ordering of hidden units by exploiting any monotonically decreasing function with unbounded support on natural numbers. With this, we do not need to fix a maximum number of neurons, which is typical of orthogonal approaches like supernetworks (White et al., 2023). As a by-product, we can achieve a straightforward trade-off between parametrization and performance by deleting the last rows/columns of weight matrices, i.e., removing the "least important" neurons from the computation. Finally, we break symmetries in the parametrization of neural networks: it is not possible anymore to obtain an equivalent neural network behavior by permuting the weight matrices, which reduces the "jostling" effect where symmetric parametrizations compete when training starts (Barber, 2012).

We test our method on MLPs for tabular data, a Convolutional Neural Network (CNN) (LeCun et al., 1989) for images, a Transformer (Vaswani et al., 2017) for text, a Recurrent Neural Network (RNN) for sequences, and a Deep Graph Network (DGN) (Micheli, 2009; Scarselli et al., 2009) for graphs, showcasing a broad scope of applicability as MLPs are ubiquitous in modern architectures. Empirical results show, as may be intuitively expected, that the width adapts to the task's difficulty with performances comparable to fixed-width baseline. We also encourage compression of the network at training time while preserving accuracy, as well as a post-hoc truncation inducing a controlled trade-off at zero additional cost. Ablations suggest that the learned width is not influenced by the starting width (under bounded activations) nor by the batch size, advocating for a potentially large reduction of the hyper-parameter configuration space.

## 2 RELATED WORK

Constructive methods dynamically learn the width of neural networks and are related in spirit to this work. The cascade correlation algorithm (Fahlman & Lebiere, 1989) alternates standard training with the creation of a new hidden unit minimizing the neural network's residual error. Similarly, the firefly network descent (Wu et al., 2020) grows the width and depth of a network every $N$ training epochs via gradient descent on a dedicated loss. Yoon et al. (2018) propose an ad-hoc algorithm for lifelong learning that grows the network by splitting and duplicating units to learn new tasks. Wu et al. (2019) alternate training the network and then splitting existing neurons into offspring with equal weights. These works mostly focus on growing the neural network; Mitchell et al. (2023) propose natural gradient-based heuristics to grow/shrink layers and hidden units of MLPs and CNNs. The main difference from our work is that we grow and shrink the network by simply computing the gradient of the loss, without relying on human-defined heuristics. The unbounded depth network of Nazaret & Blei (2022), from which we draw inspiration, learns the number of layers of neural networks. Compared to that work, we focus our attention to the number of neurons, modifying the internals of the architecture rather than instantiating a multi-output one. In particular: i) learning the width requires a different formulation of the evidence lower-bound; and ii) the importance distribution needs to be monotonically decreasing, which does not encode the right inductive bias for the depth of a neural network. We also mention Bayesian nonparametric approaches (Orbanz & Teh, 2010) that learn a potentially infinite number of clusters in an unsupervised fashion, as well as dynamic neural networks (Han et al., 2021) that condition the architecture on input properties. Finally, the work of Caron et al. (2025) investigates, in a similar spirit to this work, how to introduce decreasing rescaling of the pre-activations of neurons in the context of very wide networks; the authors provide interesting global convergence results in the Neural Tangent Kernel (NTK) regime.

**Orthogonal Methods** Neural Architecture Search (NAS) is an automated process that designs neural networks for a given task (Elsken et al., 2019b; White et al., 2023) and has been applied to different contexts (Zoph et al., 2018; Liu et al., 2019; So et al., 2019). Typically, neural network elements are added, removed, or modified based on validation performance (Elsken et al., 2019a; White et al., 2021; 2023), by means of reinforcement learning (Zoph & Le, 2016), evolutionary algorithms (Real et al., 2019), and gradient-based approaches (Liu et al., 2019). Typical NAS methods require enormous computational resources, sometimes reaching thousands of GPU days (Zoph & Le, 2016), due to the retraining of each new configuration. While recent advances on one-shot NAS models (Brock et al., 2018; Pham et al., 2018; Bender et al., 2018; Berman et al., 2020; Su et al., 2021b;a) have drastically reduced the computational costs, they mostly focus on CNNs, assume a bounded search space, and do not learn the width. As such, NAS methods are complementary to our approach. Bi-level optimization algorithms have also been used for hyper-parameter tuning (Franceschi et al., 2018), where hyper-parameters are the variables of the outer objective and the model parameters those of the inner objective. The solution sets of the inner problem are usually not available in closed form, which has been partly addressed by repeated application of (stochastic) gradient descent (Domke, 2012; Maclaurin et al., 2015; Franceschi et al., 2017). These methods are restricted to continuous hyper-parameters' optimization and cannot be applied to width optimization. Finally, pruning (Blalock et al., 2020) and distillation (Hinton et al., 2015) are two methods that reduce the size of neural networks by trading-off performances; the former deletes neural connections (Mishra et al., 2021) or entire neurons (Valerio et al., 2022; Dufort-Labbé et al., 2024), the latter trains a smaller network (student) to mimic a larger one (teacher) (Gou et al., 2021). In particular, dynamic pruning techniques can compress the network at training time (Guo et al., 2016), by applying hard or soft masks (He et al., 2018); for a comprehensive survey on pruning, please refer to (He & Xiao, 2023). It is worth mentioning the pruning strategy of Wolinski et al. (2020), which promotes learning in some neurons and penalizes others by appropriate rescaling of a potentially (infinite-dimensional) input; this work shares conceptual similarities with the our approach, though it does not consider learning the rescaling factor as a proxy for the width. Compared to most pruning approaches, our work can delete connections *and* reduce the model's memory, but also grow it indefinitely; compared to distillation, we do not necessarily need a new training to compress the network. These techniques can be easily combined with our approach, whose main goal is not compression but rather the automatic adaptation of a neural network's width.

## 3 ADAPTIVE WIDTH LEARNING

We now introduce Adaptive Width Neural Networks (AWN), a probabilistic framework that maximizes a simple variational objective via backpropagation over a neural network's parameters.

We are given a dataset of $N$ *i.i.d.* samples, with input $x \in \mathbb{R}^F$, $F \in \mathbb{N}^+$ and target $y$ whose domain depends on whether the task is regression or classification. For samples $X \in \mathbb{R}^{N \times F}$ and targets $Y$, the learning objective is to maximize the log-likelihood

$$\log p(Y|X) = \log \prod_{i=1}^{N} p(y_i|x_i) = \sum_{i=1}^{N} \log p(y_i|x_i) \tag{1}$$

with respect to the learnable parameters of $p(y|x)$.

We define $p(y|x)$ according to the graphical model of Figure 1 (left), to learn a neural network with **unbounded width** for each hidden layer $\ell$. To do so, we assume the existence of an **infinite** sequence of *i.i.d.* latent variables $\boldsymbol{\theta}_\ell = \{\theta_{\ell n}\}_{n=1}^{\infty}$, where $\theta_{\ell n}$ is a multivariate variable over the learnable weights of neuron $n$ at layer $\ell$. However, since working with an infinite-width layer is not possible in practice, we also introduce a latent variable $\lambda_\ell$ that samples how many neurons to use for layer $\ell$. That is, it *truncates an infinite width to a finite value* so that we can feasibly perform inference with the neural network. For a neural network of $L$ layers, we define $\boldsymbol{\theta} = \{\boldsymbol{\theta}_\ell\}_{\ell=1}^{L}$ and $\boldsymbol{\lambda} = \{\lambda_\ell\}_{\ell=1}^{L}$, assuming independence across layers. Therefore, by marginalization one can write $p(Y|X) = \int p(Y, \boldsymbol{\lambda}, \boldsymbol{\theta}|X) d\boldsymbol{\lambda} d\boldsymbol{\theta}$.

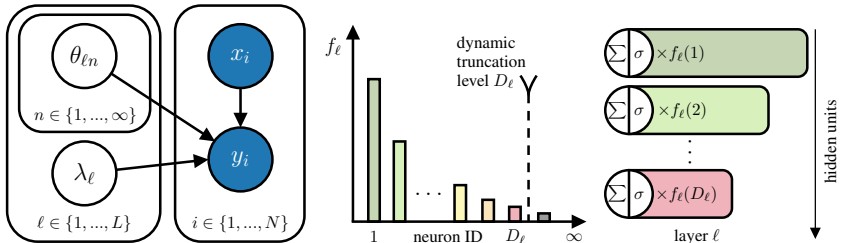

Figure 1: *(Left)* The graphical model of AWN, with dark observable random variables and white latent ones. *(Middle)* The distribution $f_\ell$ over hidden units' importance at layer $\ell$ is parametrized by $\lambda_\ell$. The width of layer $\ell$ is chosen as the quantile function of the distribution $f_\ell$ evaluated at $k$ and denoted by $D_\ell$. *(Right)* The hidden units' activations at layer $\ell$ are rescaled by their importance.

We then decompose the joint distribution using the independence assumptions of the graphical model:

$$p(Y, \boldsymbol{\lambda}, \boldsymbol{\theta}|X) = \prod_{i=1}^{N} p(y_i, \boldsymbol{\lambda}, \boldsymbol{\theta}|x_i) \qquad p(y_i, \boldsymbol{\lambda}, \boldsymbol{\theta}|x_i) = p(y_i|\boldsymbol{\lambda}, \boldsymbol{\theta}, x_i)p(\boldsymbol{\lambda})p(\boldsymbol{\theta}) \tag{2}$$

$$p(\boldsymbol{\lambda}) = \prod_{\ell=1}^{L} p(\lambda_\ell) = \prod_{\ell=1}^{L} \mathcal{N}(\lambda_\ell; \mu_\ell^\lambda, \sigma_\ell^\lambda) \quad p(\boldsymbol{\theta}) = \prod_{\ell=1}^{L}\prod_{n=1}^{\infty} p(\theta_{\ell n}) = \prod_{\ell=1}^{L}\prod_{n=1}^{\infty} \mathcal{N}(\theta_{\ell n}; \mathbf{0}, \mathrm{diag}(\sigma_\ell^\theta)) \tag{3}$$

$$p(y_i|\boldsymbol{\lambda}, \boldsymbol{\theta}, x_i) = \text{Neural Network as will be introduced in Section 3.1} \tag{4}$$

where $\sigma_\ell^\theta, \mu_\ell^\lambda, \sigma_\ell^\lambda$ are hyper-parameters. For simplicity, we assumed a Gaussian prior for $\boldsymbol{\lambda}$, but please note that, when strict positivity is required, one should formally turn to Folded Normal priors. The neural network is parametrized by realizations $\boldsymbol{\lambda} \sim p(\boldsymbol{\lambda}), \boldsymbol{\theta} \sim \boldsymbol{\theta}$ – it relies on a finite number of neurons as detailed later and in Section 3.1 – and it outputs either class probabilities (classification) or the mean of a Gaussian distribution (regression) to parametrize $p(y_i|\boldsymbol{\lambda}, \boldsymbol{\theta}, x_i)$ depending on the task. Maximizing Equation (1), however, requires computing the evidence $\int p(Y, \boldsymbol{\lambda}, \boldsymbol{\theta}|X)d\boldsymbol{\lambda}d\boldsymbol{\theta}$, which is intractable. Therefore, we turn to mean-field variational inference (Jordan et al., 1999; Blei et al., 2017) to maximize an expected lower bound (ELBO) instead. This requires to define a variational distribution over the latent variables $q(\boldsymbol{\lambda}, \boldsymbol{\theta})$ and re-phrase the objective as:

$$\log p(Y|X) \geq \mathbb{E}_{q(\boldsymbol{\lambda}, \boldsymbol{\theta})}\left[\log \frac{p(Y, \boldsymbol{\lambda}, \boldsymbol{\theta}|X)}{q(\boldsymbol{\lambda}, \boldsymbol{\theta})}\right], \tag{5}$$

where $q(\boldsymbol{\lambda}, \boldsymbol{\theta})$ is parametrized by learnable *variational* parameters. Before continuing, we define the **truncated width** $D_\ell$, that is the finite number of neurons at layer $\ell$, as the quantile function evaluated at $k$, with $k$ a hyper-parameter, of a distribution[1] $f_\ell$ with infinite support over $\mathbb{N}^+$ and parametrized by $\lambda_\ell$; Appendix A provides desirable properties of $f_\ell$ in a similar vein to Nazaret & Blei (2022). In other words, we find the integer such that the cumulative mass function of $f_\ell$ takes value $k$, and that integer $D_\ell$ is the truncated width of layer $\ell$. We implement $f_\ell$ as a **discretized exponential distribution** adhering to Def. A.2, following the discretization strategy of Roy (2003): For every $x \in \mathbb{N}^+$, the discretized distribution relies on the exponential's cumulative distribution function:

$$f_\ell(x; \lambda_\ell) = (1 - e^{-\lambda_\ell(x+1)}) - (1 - e^{-\lambda_\ell x}). \tag{6}$$

We choose the exponential because it is a **monotonically decreasing** function and allows us to impose an ordering of importance among neurons, as detailed in Section 3.1.
Then, we can factorize the variational distribution $q(\boldsymbol{\lambda}, \boldsymbol{\theta})$ into:

$$q(\boldsymbol{\lambda}, \boldsymbol{\theta}) = q(\boldsymbol{\lambda})q(\boldsymbol{\theta}|\boldsymbol{\lambda}) \qquad q(\boldsymbol{\lambda}) = \prod_{\ell=1}^{L} q(\lambda_\ell) = \prod_{\ell=1}^{L} \mathcal{N}(\lambda_\ell; \nu_\ell, 1) \tag{7}$$

$$q(\boldsymbol{\theta}|\boldsymbol{\lambda}) = \prod_{\ell=1}^{L}\prod_{n=1}^{D_\ell} q(\theta_{\ell n}) \prod_{n'=D_\ell+1}^{\infty} p(\theta_{\ell n'}) \qquad q(\theta_{\ell n}) = \mathcal{N}(\theta_{\ell n}; \rho_{\ell n}, \mathbf{I}). \tag{8}$$

---

[1]General functions are allowed if a threshold can be computed.

Here, $\nu_\ell, \rho_{\ell n}$ are learnable variational parameters and, as before, we define $\boldsymbol{\rho}_\ell = \{\rho_{\ell n}\}_{n=1}^{D_\ell}$, $\boldsymbol{\rho} = \{\boldsymbol{\rho}_\ell\}_{\ell=1}^L$ and $\boldsymbol{\nu} = \{\nu_\ell\}_{\ell=1}^L$. Note that the set of variational parameters is **finite** as it depends on $D_\ell$.

By expanding Equation 5 using the above definitions and approximating the expectations at the first order, $i.e.$, $\mathbb{E}_{q(\boldsymbol{\lambda})}[f(\boldsymbol{\lambda})]=f(\boldsymbol{\nu})$ and $\mathbb{E}_{q(\boldsymbol{\theta}|\boldsymbol{\lambda})}[f(\boldsymbol{\theta})] = f(\boldsymbol{\rho})$ as in Nazaret & Blei (2022), we obtain the final form of the ELBO (the full derivation is in Appendix C):

$$\sum_\ell^L \log \frac{p(\nu_\ell; \mu_\ell^\lambda, \sigma_\ell^\lambda)}{q(\nu_\ell; \nu_\ell)} + \sum_\ell^L \sum_{n=1}^{D_\ell} \log \frac{p(\rho_{\ell n}; \sigma_\ell^\theta)}{q(\rho_{\ell n}; \rho_{\ell n})} + \sum_{i=1}^N \log p(y_i | \boldsymbol{\lambda}{=}\boldsymbol{\nu}, \boldsymbol{\theta}{=}\boldsymbol{\rho}, x_i), \qquad (9)$$

where distributions' parameters are made explicit to distinguish them. The first two terms in the ELBO regularize the width of the layers and the magnitude of the parameters when priors are informative, whereas the third term accounts for the predictive performance.

In practice, the finite variational parameters $\boldsymbol{\nu}, \boldsymbol{\rho}$ are those used to parametrize the neural network rather than sampling $\boldsymbol{\lambda}, \boldsymbol{\theta}$, which enables easy optimization via backpropagation. Maximizing Equation (9) will update each variational parameter $\nu_\ell$, which in turn will change the value of $D_\ell$ **during training**. If $D_\ell$ increases we initialize new neurons and draw their weights from a standard normal distribution, otherwise we discard the weights of the extra neurons. When implementing mini-batch training, the predictive loss needs to be rescaled by $N/M$, where $M$ is the mini-batch size. From a Bayesian perspective, this is necessary as regularizers should weigh less if we have more data.

Compared to a fixed-width network with weight decay, we need to choose the priors' values of $\mu_\ell^\lambda, \sigma_\ell^\lambda$, as well as initialize the learnable $\nu_\ell$. The latter can be initially set to same value across layers since they can freely adapt later, or it can be sampled from the prior $p(\lambda_\ell)$. Therefore, we have two more hyper-parameters compared to the fixed-width network, but we make some considerations: **i)** it is always possible to use an uninformative prior over $\lambda_\ell$, removing the extra hyper-parameters letting the model freely adapt the width of each layer (as is typical of frequentist approaches); **ii)** the choice of higher level of hyper-parameters is known to be less stringent than that of hyper-parameters themselves (Goel & Degroot, 1981; Bernardo & Smith, 2009), so we do not need to explore many values of $\mu_\ell^\lambda$ and $\sigma_\ell^\lambda$; **iii)** our experiments suggest that AWN can converge to similar widths regardless of the starting point $\nu_\ell$, so that we may just need to perform model selection over one/two sensible initial values; **iv)** the more data, the less the priors will matter.

### 3.1 Imposing a Soft Ordering on Neurons' Importance

Now that the learning objective has been formalized, the missing ingredient is the definition of the neural network $p(y_i | \boldsymbol{\lambda}{=}\boldsymbol{\nu}, \boldsymbol{\theta}{=}\boldsymbol{\rho}, x_i)$ of Equation 4 as a modified MLP. Compared to a standard MLP, we use the variational parameters $\boldsymbol{\nu}$ that affect the truncation width at each hidden layer, whereas $\boldsymbol{\rho}$ are the weights. We choose a monotonically decreasing function $f_\ell$, thus when a new neuron is added its relative importance is low and will not drastically impact the network output and hidden activations. In other words, we impose a soft ordering of importance among neurons.
We simply modify the classical activation $h_j^\ell$ of a hidden neuron $j$ at layer $\ell$ as

$$h_j^\ell = \sigma \left( \sum_{k=1}^{D_{\ell-1}} w_{jk}^\ell h_k^{\ell-1} \right) f_\ell(j; \nu_\ell), \qquad (10)$$

where $D_{\ell-1}$ is the truncated width of the previous layer, $\sigma$ is a non-linear activation function and $w_{jk}^\ell \in \rho_{\ell j}$. That is, we rescale the activation of each neuron $k$ by its "importance" $f_\ell(j; \nu_\ell)$. Note that the bias parameter is part of the weight vector as usual.

It is easy to see that, in theory, the optimization algorithm could rescale the weights of the next layer by a factor $1/f_\ell(j; \nu_\ell)$ to compensate for the term $f_\ell(j; \nu_\ell)$. This could lead to a degenerate situation where the activations of the first neurons are small relative to the others, thus breaking the soft-ordering and potentially wasting neurons. There are two strategies to address this undesirable effect. The first is to regularize the magnitude of the weights thanks to the prior $p(\theta_{\ell+1n})$, so that it may be difficult to compensate for the least important neurons that have a high $1/f_\ell(j)$. The second and less obvious strategy is to prevent the units' activations of the current layer to compensate for high values by bounding their range, e.g., using a ReLU6 or tanh activation (Sandler et al., 2018). We apply both strategies to our experiments, although we noted that they do not seem strictly necessary in practice.

## 3.2 Rescaled Weight Initialization for Deep AWN

Rescaling the activations of hidden units using Equation 10 causes activations of deeper layers to quickly decay to zero for an AWN MLP with ReLU nonlinearity initialized using the well known Kaiming scheme (He et al., 2015). This affects convergence since gradients get close to zero and it becomes slow to train deep AWN MLPs. We therefore derive a rescaled Kaiming weight that guarantees that the variance of activation across layers is constant at initialization.

**Theorem 3.1.** *Consider an MLP with activations as in Equation 10 and ReLU nonlinearity. At initialization, given* $\alpha_j^\ell = \sigma\left(\sum_{k=1}^{D_{\ell-1}} w_{jk}^\ell h_k^{\ell-1}\right), \mathrm{Var}[w_{jk}^\ell] = \frac{2}{\sum_{j=1}^{D_{\ell-1}} f_\ell^2(j)} \Rightarrow \mathrm{Var}[\alpha_j^\ell] \approx \mathrm{Var}[\alpha_j^{\ell-1}].$

*Proof.* See Appendix E for an extended proof. □

The extended proof shows there is a connection between the variance of activations and the variance of gradients. In particular, the sufficient conditions over $\mathrm{Var}[w_{j*}^\ell]$ are identical if one initializes $\nu_\ell$ in the same way for all layers. Therefore, if we initialize weights from a Gaussian distribution with standard deviation $\frac{\sqrt{2}}{\sqrt{\sum_{j=1}^{D_{\ell-1}} f_\ell^2(j)}}$, we guarantee that at initialization the variance of the deep network's gradients will be constant. The effect of the new initialization (dubbed "Kaiming+") can be seen in Figure 7, where the distribution of activation values at initialization does not collapse in subsequent layers. This change drastically impacts overall convergence on the SpiralHard synthetic dataset (described in Section 4), where it appears it would be otherwise hard to converge using a standard Kaiming initialization.

Algorithm 1 summarizes the main changes to the training procedure, namely the new initialization and the update of the model's truncated width at each training step.

---

**Algorithm 1** AWN Training Procedure

1: **Input:** Dataset $\mathcal{D}$, initialized AWN model $\mathcal{M}$ (Section 3.2)
2: **Output:** Trained AWN Model $\mathcal{M}$
3: **for** each training epoch **do**
4:     **for** batch in $\mathcal{D}$ **do**
5:         update_width($\mathcal{M}$)
6:         $\hat{y} \leftarrow \mathcal{M}(\text{batch})$
7:         $\text{loss} \leftarrow \text{ELBO}(\mathcal{M}, \text{batch}, \hat{y})$     // Eq. 9
8:         $\mathcal{M} \leftarrow \text{backpropagation}(\mathcal{M}, \text{loss})$
9: **function** update_width($\mathcal{M}$):
10:     **for** layer $\ell$ in $\mathcal{M}$.hidden_layers **do**
11:         $D_\ell \leftarrow$ quantile function of $f_\ell(\cdot; \nu_\ell)$ evaluated at $k$
12:         Use $D_\ell$ to update $\boldsymbol{\rho}_\ell, \boldsymbol{\rho}_{\ell+1}$    // add/remove neurons

---

## 3.3 Future Directions and Limitations

MLPs are ubiquitous in modern deep architectures. They are used at the very end of CNNs, in each Transformer layer, and they process messages coming from neighbors in DGNs. Our experiments focus on MLPs to showcase AWN's broad applicability, but there are many other scenarios where one can apply AWN's principles. For instance, one could impose a soft ordering of importance on CNNs' filters at each layer, therefore learning the number of filters during training. However, doing so would require a distinct formalization, which is why this lies beyond the scope of the present work.

From a more theoretical perspective, we believe one could draw connections between our technique and the Information Bottleneck principle (Tishby et al., 2000), which seeks maximally representative (i.e., performance) and compact representations (e.g., width). In addition, we could try to revisit the theoretical results of Caron et al. (2025) in the NTK regime to better understand convergence properties of adaptive-width neural networks. since both are based on asymmetric rescaling of activations.

## 4 Experiments and Setup

The purpose of the empirical analysis is not to claim AWN is generally better than the fixed-width baseline. Rather, we demonstrate how AWN overcomes the problem of fixing the number of neurons by learning it end-to-end, thus reducing the amount of hyper-parameter configurations to test. As such, due to the nature of this work and similarly to Mitchell et al. (2023), we use the remaining space to thoroughly study the behavior of AWN, so that it becomes clear how to use it in practice. We first quantitatively verify that AWN does not harm the performance compared to baseline models and compare the chosen width by means of grid-search model selection with the learned width of AWN.

Second, we check that AWN chooses a larger width for harder tasks, which can be seen as increasing the hypotheses space until the neural network finds a good path to convergence. Third, we verify that convergence speed is not significantly altered by AWN, so that the main limitation lies in the extra overhead for adapting the network at each training step. As a sanity check, we study conditions under which AWN's learned width does not seem to depend on starting hyper-parameters, so that their choice does not matter much. In addition, we analyze other practical advantages of training a neural network under the AWN framework: the ability to compress information during training or post training, and the resulting trade-offs. Finally, we analyze the impact of different families of functions $f_\ell(j; \nu_\ell)$ compared to the exponential mainly used in this work. Further analyses are in the Appendix.

We compare baselines that undergo proper hyper-parameter tuning (called "Fixed") against its AWN version, where we replace any fixed MLP with an adaptive one. First, we train an MLP on 3 synthetic tabular tasks of increasing binary classification difficulty, namely a double moon, a spiral, and a double spiral that we call SpiralHard. A stratified hold-out split of 70% training/10% validation/20% test for risk assessment is chosen at random for these datasets. Then, we consider 3 larger real-world tabular tasks with higher feature dimensionality, namely pol, MiniBooNE, and credit card clients (Beyazit et al., 2023). Similarly, we consider a ResNet-20 (He et al., 2016) trained on 3 image classification tasks, namely MNIST (LeCun, 1998), CIFAR10, and CIFAR100 (Krizhevsky, 2009), where data splits and preprocessing are taken from the original paper and AWN is applied to the downstream classifier. In the sequential domain, we implemented a basic adaptive Recurrent Neural Network (RNN) and evaluated on the PMNIST dataset (Zenke et al., 2017) with the same data split as MNIST. In the graph domain, we train a Graph Isomorphism Network (Xu et al., 2019) on the NCI1 and REDDIT-B classification tasks, where topological information matters, using the same split and evaluation setup of Errica et al. (2020). Here, the first 1 hidden layer MLP as well as the one used in each graph convolutional layer are replaced by adaptive AWN versions. On all these tasks, the metric of interest is the accuracy. Finally, for the textual domain we train a Transformer architecture (Vaswani et al., 2017) on the Multi30k English-German translation task (Elliott et al., 2016), using a pretrained GPT-2 Tokenizer, and we evaluate the cross-entropy loss over the translated words. On tabular, image, and text-based tasks, an internal validation set (10%) for model selection is extracted from the union of outer training and validation sets, and the best configuration chosen according to the internal validation set is retrained 10 times on the outer train/validation/test splits, averaging test performances after an early stopping strategy on the validation set. Due to space reasons, we report datasets statistics and the hyper-parameter tried for the fixed and AWN versions in Appendix F and G, respectively[2]. We ran the experiments on a server with 64 cores, 1.5TB of RAM, and 4 NVIDIA A40 with 48GB of memory.

## 5 RESULTS

We begin by discussing the quantitative results of our experiments: Table 1 reports means and standard deviations across the 10 final training runs. In terms of performance, we observe that AWN is more stable or accurate than a fixed MLP on tabular and sequential datasets; all other things being equal, it seems that using more neurons and their soft ordering are the main contributing factors to these improvements. On the image datasets, performances of AWN are comparable to those of the fixed baseline but for CIFAR100, due to an unlucky run that did not converge. There, AWN learns a smaller total width compared to grid search.

Results on graph datasets are interesting in two respects: First, the performance on REDDIT-B is significantly improved by AWN both in terms of average performance and stability of results; second, and akin to PMNIST, the total learned width is significantly higher than those tried in Xu et al. (2019); Errica et al. (2020), meaning a biased choice of possible widths had a profound influence on risk estimation for DGN models (i.e., GIN). This result makes it evident that it is important to let the network decide how many neurons are necessary to solve the task. Appendix H shows what happens when we retrain Fixed baselines using the total width as the width of each layer.

Finally, the results on the Multi30k show that the AWN Transformer learns to use 200x parameters less than the fixed Transformer for the feed-forward networks, achieving a statistically comparable

---

[2]Code to reproduce results is available at `https://github.com/nec-research/Adaptive-Width-Neural-Networks`.

Table 1: Performances and total width of MLP layers for the fixed and AWN versions of the various models used. The exact width chosen by model selection on the graph datasets is unknown since we report published results. "Linear" means the chosen downstream classifier is a linear model.

| | Fixed | | AWN | | Width (Fixed) | Width (AWN) | |
|---|---|---|---|---|---|---|---|
| | Mean | (Std) | Mean | (Std) | | Mean | (Std) |
| DoubleMoon | 100.0 | (0.0) | 100.0 | (0.0) | 8 | 8.1 | (2.8) |
| Spiral | 99.5 | (0.5) | 99.8 | (0.1) | 16 | 65.9 | (8.7) |
| SpiralHard | 98.0 | (2.0) | 100.0 | (0.0) | 32 | 227.4 | (32.4) |
| pol | 99.3 | (0.2) | 99.2 | (0.1) | 48 | 84 | (11.0) |
| MiniBooNE | 92.9 | (0.3) | 93.2 | (0.1) | 32 | 53 | (11.1) |
| credit card | 81.6 | (0.1) | 81.8 | (0.1) | 16 | 51 | (12.0) |
| PMNIST | 91.1 | (0.4) | 95.7 | (0.2) | 24 | 806.3 | (44.5) |
| MNIST | 99.6 | (0.1) | 99.7 | (0.0) | Linear | 19.4 | (4.8) |
| CIFAR10 | 91.4 | (0.2) | 91.4 | (0.2) | Linear | 80.1 | (12.4) |
| CIFAR100 | 66.5 | (0.4) | 63.1 | (4.0) | 256 | 161.9 | (57.8) |
| NCI1 | 80.0 | (1.4) | 80.0 | (1.1) | (96-320) | 731.3 | (128.2) |
| REDDIT-B | 87.0 | (4.4) | 90.2 | (1.3) | (96-320) | 793.6 | (574.0) |
| Multi30k ($\downarrow$) | 1.43 | (0.4) | 1.51 | (0.2) | 24576 | 123.2 | (187.9) |

test loss. This result is appealing when read through the lenses of modern deep learning, as the power required by some neural networks such as Large Language Models (Brown et al., 2020) is so high that cannot be afforded by most institutions, and it demands future investigations.

**Adaptation to Task Difficulty and Convergence** Intuitively, one would expect that AWN learned larger widths for more difficult tasks. This is indeed what happens on the tabular datasets (and image datasets, see Appendix I) where some tasks are clearly harder than others. Figure 2 (left) shows that, given the same starting width per layer, the learned number of neurons grows according to the task's difficulty. It is also interesting to observe that the total width for a multi-layer MLP on SpiralHard is significantly lower than that achieved by a single-layer MLP, which is consistent with the circuit complexity theory arguments put forward in Bengio et al. (2006); Mhaskar et al. (2017). It also appears that convergence is not affected by the introduction of AWN, as investigated in Figure 2 (right), which was not obvious considering the parametrization constraints encouraged by the rescaling of neurons' activations.

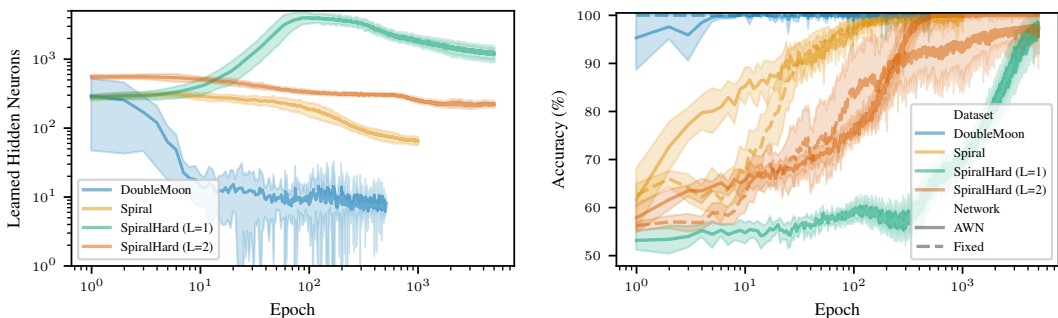

Figure 2: (Left) The learned width adapts to the increasing difficulty of the task, from the DoubleMoon to SpiralHard. (Right) AWN reaches perfect test accuracy with a comparable amount of epochs on DoubleMoon and Spiral, while it converges faster on SpiralHard.

**Training Stability Analysis** To support our argument that AWN can reduce the time spent performing hyper-parameter selection, we check whether AWN learns a consistent amount of neurons across different training runs and hyper-parameter choices. Figure 3 reports the impact of the batch size and starting width averaged across the different configurations tried during model selection. Smaller batch sizes cause more instability, but in the long run we observe convergence to a similar

width. Convergence with respect to different rates holds, instead, for the bounded ReLU6 activation; Appendix K shows that unbounded activations may cause the network to converge more slowly to the same width, which is in accord with the considerations about counterbalancing the rescaling effect of Section 3.1. Therefore, whenever possible, we recommend using bounded activations.

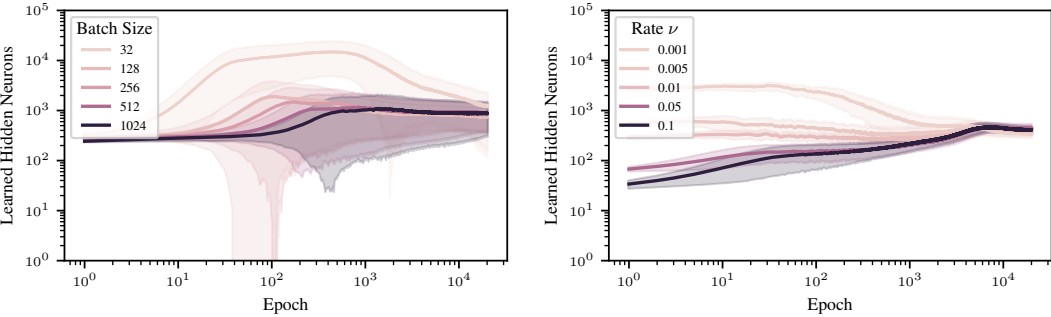

Figure 3: Training converges to similar widths on SpiralHard for different batch sizes (left) and starting rates $\nu$, but the latter seems to require a bounded nonlinearity such as ReLU6 to converge in a reasonable amount of epochs (right).

**Online Network Compression via Regularization**   So far, we have used an uninformative prior $p(\lambda)$ over the neural networks' width. We demonstrate the effect of an informative prior by performing a width-annealing experiment on the SpiralHard dataset. We set an uninformative $p(\boldsymbol{\theta})$ and ReLU6 nonlinearity. At epoch 1000, we introduce $p(\lambda_\ell) = \mathcal{N}(\lambda_\ell; 0.05, 1)$, and gradually anneal the standard deviation up to $0.1$ at epoch 2500. Figure 4 shows that the width of the network reduces from approximately 800 neurons to 300 without any test performance degradation. We hypothesize that the least important neurons carry negligible information, therefore they can be safely removed without drastic changes in the output of the model. *This technique might be particularly useful to compress large models with billions of parameters.*

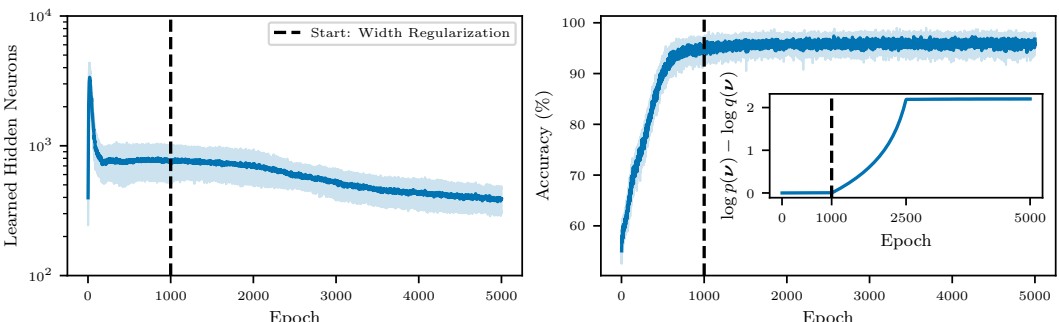

Figure 4: It is possible to regularize the width at training time by increasing the magnitude of the loss term $\log \frac{p(\boldsymbol{\nu})}{q(\boldsymbol{\nu})}$. The total width is reduced by more than 50% (left) while preserving accuracy (right). The inset plot refers to the loss term that AWN tries to maximize.

**Post-hoc Truncation Achieves a Trade-off between Performance and Compute Resources**   To further investigate the consequences of imposing a soft ordering among neurons, we show that it is possible to perform straightforward post-training truncation while still controlling the trade-off between performance and network size. Figure 5 shows an example for an MLP on the Spiral dataset, where the range of activation values (Equation 10) computed for all samples follows an exponential curve (right). Intuitively, removing the last neurons may have a negligible performance impact at the beginning and a drastic one as few neurons remain. This is what happens, where we are able to cut an MLP with hidden width 83 by 30% without loss of accuracy, after which a smooth degradation happens. If one accepts such a trade-off, this technique may be used to "distill" a trained neural network at virtually zero additional cost while reducing the memory requirements. Note that

truncation heuristics that are either random or based on the magnitude of neurons' activations (i.e., excluding the rescaling term) do not perform as well.

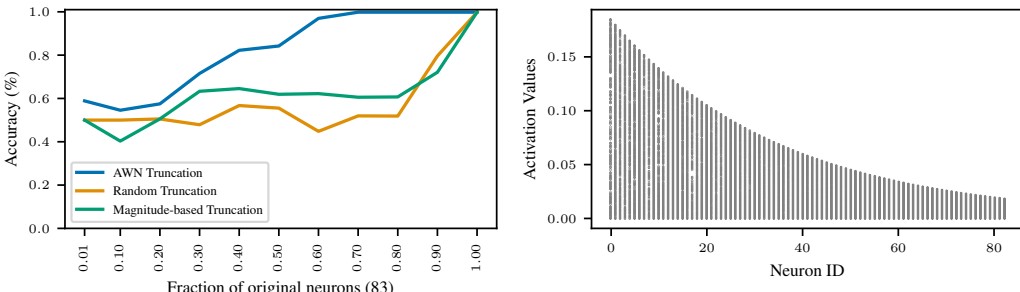

Figure 5: (Left) Thanks to the soft ordering imposed on the neurons, one can also truncate the neural network *after training* by simply removing the last neurons. (Right) The distribution of neurons' activations for all Spiral test samples follows an exponential-like curve.

**Impact of Different Functions** $f_\ell(j; \nu_\ell)$    We conclude our main analyses with an ablation study on the use of different decreasing functions $f_\ell(j; \nu_\ell)$, considering the SpiralHard dataset, to understand how different functions have different properties. We tested: i) a Power Law distribution $f_\ell(j; \nu_\ell, \alpha) = Z(j + j_{sat})^{-\nu_\ell}$, with $Z$ being a normalizing term, initial $\nu_\ell \in [2., 3.]$ and low degree saturation $j_{sat}$ chosen between $[0., 2.]$; ii) a sigmoid-like function $f_\ell(j; \nu_\ell, b) = 1 - \sigma(j - \nu_\ell)$, where $\sigma$ is the sigmoidal activation, $\nu_\ell \in [128, 256]$ is the transition point.

The results are reported in Table 2. When averaged over all possible hyper-parameter configurations, there is no significant deviation in performance between distribution, and there is always at least one configuration attaining the best performance. However, the Power Law distribution introduces more neurons than the exponential due to its long-tail. Instead, the Sigmoidal function tends to allocates less neurons, but the drawback is the loss of the ability to truncate the network after training, since the importance of neurons is very close to 1 before the transition point and the transition itself, for the chosen fixed parameters, is relatively sharp. The choice of the function family $f_\ell(j; \nu_\ell)$ remains an interesting direction of further exploration, given that each family has different properties that might be more or less suitable depending on the use cases.

Table 2: Analysis of the impact of different families of importance functions on SpiralHard validation performance and learned width, averaged across different hyper-parameter configurations.

| $f_\ell(j; \nu_\ell)$ | **Mean Accuracy** | **Max Accuracy** | **Mean Total Width** |
|---|---|---|---|
| Exponential | 80.27 (19.9) | 100.00 | 954.4 (1083.9) |
| Power Law | 81.82 (16.6) | 100.00 | 2952.4 (3371.6) |
| Sigmoidal | 76.85 (18.3) | 100.00 | 426.8 (268.4) |

## 6    CONCLUSIONS

We introduced a new methodology to learn an unbounded width of neural network layers within a single training, by imposing a soft ordering of importance among neurons. Our approach requires very few changes to the architecture, adapts the width to the task's difficulty, and does not impact negatively convergence. We showed stability of convergence to similar widths under bounded activations for different hyper-parameters configurations, advocating for a practical reduction of the width's search space. A by-product of neurons' ordering is the ability to easily compress the network during or after training, which is relevant in the context of foundational models trained on large data, which are believed to require billions of parameters. Finally, we have tested AWN on different models and data domains to prove its broad scope of applicability: a Transformer architecture achieved a similar loss with 200x less parameters.

## 7 REPRODUCIBILITY STATEMENT

The code provided in the supplementary material relies on libraries for automatic experimentation, which enforce reproducibility of the results. These libraries require that the user specifies every detail, from data splitting strategies to hyper-parameter selection and final evaluations, and subsequently the experiment is automatized and results are provided.

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

## A  TRUNCATED DISTRIBUTIONS' NOTIONS

Since we were mostly inspired by the work of Nazaret & Blei (2022), we complement the main text with an introduction to truncated distributions. In particular, by truncating a distribution at its quantile function evaluated at $k$ (see Appendix B for a visual explanation), the support of the distribution becomes finite and countable, hence we can compute expectations in a finite number of steps. A truncated distribution should ideally satisfy some requirements defined in Nazaret & Blei (2022) and illustrated below.

**Definition A.1** (Nazaret & Blei (2022)). A variational family $Q = q(x; \boldsymbol{\omega})$ over $\mathbb{N}^+$ is *unbounded* with *connected* and *bounded* members if

1. $\forall q \in Q, \text{support}(q)$ is bounded

2. $\forall L \in \mathbb{N}^+, \exists q \in Q$ such that $L \in \text{argmax}(q)$

3. Each parameter in the set $\boldsymbol{\omega}$ is a continuous variable.

The first condition allows us to compute the expectation over any $q \in Q$ in finite time, condition 2 ensures there is a parametrization that assigns enough probability mass to each point in the support of $q$, and condition 3 is necessary for learning via backpropagation.

An important distinction with Nazaret & Blei (2022) is that **we do not need nor want condition 2 to be satisfied**. As a matter of fact, we want to enforce a strict ordering of neurons where the first is always the most important one. Condition 2 is useful, as done in Nazaret & Blei (2022), to assign enough 'importance" to a specific layer of an adaptive-depth architecture, while previous layers are still functional to the final result. In the context of adaptive-width networks, however, this mechanism may leave a lot of neurons completely unutilized, letting the network grow indefinitely. That is why we require that each distribution $f_\ell$ is monotonically decreasing and all possible widths should have non-zero importance. We change the definition for the distributions we are interested in as follows:

**Definition A.2.** A family of distributions $Q = f(x; \boldsymbol{\lambda})$ over $\mathbb{N}^+$ is *unbounded* with *connected*, *bounded*, and *decreasing* members if

1. $\forall f \in Q, \text{support}(f)$ is bounded

2. $\forall f \in Q$, $f$ is monotonically decreasing and $\forall x \in \mathbb{N}^+, f(x) > 0$

3. Each parameter in the set $\boldsymbol{\lambda}$ is a continuous variable.

## B  HOW TO COMPUTE $D_\ell$ IN PRACTICE

To aid the understanding of how one computes $D_\ell$, we provide a visualization of the exponential distribution, its discretized version (Equation 6) and the quantile function evaluated at $k = 0.9$ in Figure 6 below.

Note that typically, newly introduced neurons are multiplied by a very small importance, and their random weights are sampled from a standard Gaussian. This makes their initial contribution very small, so it does not drastically alter the loss/gradients/convergence. From a backpropagation viewpoint, the reason why new neurons are introduced is that previous neurons need to become relatively more important. So, when the chosen importance distributions grow too slowly, as in the case of a Power law family of distributions, this can slow down convergence because it takes a long time to make the last neurons more important, and in addition a large tail or unimportant neurons is added if the chosen quantile is too high. The exponential distribution, instead, worked pretty well without having to tune it extensively.

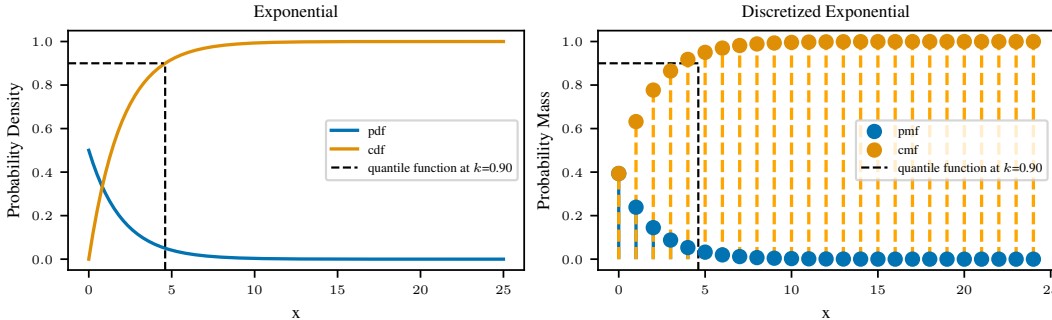

Figure 6: On the left, we provide a visualization of the exponential density with $\lambda = 0.5$, its cumulative density, and the quantile function evaluated at $k = 0.9$, which returns a value close to $5$. On the right, we show the discretized exponential and its cumulative mass function; we can see that the quantile function of the continuous density provides an upper bound for the natural where the cumulative mass function is greater than $k = 0.9$. Therefore, one can safety use the ceiling of the output of the quantile function to determine the width $D_\ell$ to use for a given value of $\lambda$.

## C    FULL ELBO DERIVATION

The full ELBO derivation is as follows:

$$\mathbb{E}_{q(\boldsymbol{\lambda},\boldsymbol{\theta})}\left[\log \frac{p(Y,\boldsymbol{\lambda},\boldsymbol{\theta}|X)}{q(\boldsymbol{\lambda},\boldsymbol{\theta})}\right] = \tag{11}$$

$$\mathbb{E}_{q(\boldsymbol{\lambda})}\mathbb{E}_{q(\boldsymbol{\theta}|\boldsymbol{\lambda})}[\log p(Y,\boldsymbol{\lambda},\boldsymbol{\theta}|X) - \log q(\boldsymbol{\lambda},\boldsymbol{\theta})] = \tag{12}$$

$$\mathbb{E}_{q(\boldsymbol{\lambda})}\mathbb{E}_{q(\boldsymbol{\theta}|\boldsymbol{\lambda})}[\log(p(Y|\boldsymbol{\lambda},\boldsymbol{\theta},X)p(\boldsymbol{\theta})p(\boldsymbol{\lambda})) - \log(q(\boldsymbol{\lambda})q(\boldsymbol{\theta}|\boldsymbol{\lambda}))] \tag{13}$$

where we factorized probabilities according to the independence assumptions of the graphical model. Then

$$\mathbb{E}_{q(\boldsymbol{\lambda})}\mathbb{E}_{q(\boldsymbol{\theta}|\boldsymbol{\lambda})}[\log p(Y|\boldsymbol{\lambda},\boldsymbol{\theta},X) + \log p(\boldsymbol{\lambda}) + \log p(\boldsymbol{\theta}) - \log q(\boldsymbol{\lambda}) - \log q(\boldsymbol{\theta}|\boldsymbol{\lambda})] = \tag{14}$$

$$\mathbb{E}_{q(\boldsymbol{\lambda})}\mathbb{E}_{q(\boldsymbol{\theta}|\boldsymbol{\lambda})}[\log \frac{p(\boldsymbol{\lambda})}{q(\boldsymbol{\lambda})} + \log \frac{p(\boldsymbol{\theta})}{q(\boldsymbol{\theta}|\boldsymbol{\lambda})} + \log p(Y|\boldsymbol{\lambda},\boldsymbol{\theta},X)] \approx \tag{15}$$

$$\log \frac{p(\boldsymbol{\nu})}{q(\boldsymbol{\nu})} + \log \frac{p(\boldsymbol{\rho})}{q(\boldsymbol{\rho}|\boldsymbol{\nu})} + \log p(Y|\boldsymbol{\nu},\boldsymbol{\rho},X)] \tag{16}$$

where we use the first-order approximation in the last step. Note that when computing $\frac{p(\theta)}{q(\theta|\lambda)}$, the products from $D_{\ell+1}$ to infinity cancel out. Equation 13 follows by simply applying definitions, the iid assumption $p(Y|\boldsymbol{\nu},\boldsymbol{\rho},X) = \prod_i^N p(y_i|\boldsymbol{\nu},\boldsymbol{\rho},x_i)$, and by transforming products into summations using the logarithm.

An important note about the first-order (Taylor) approximation $E_{q(\boldsymbol{\lambda};\boldsymbol{\nu})q(\boldsymbol{\theta}|\boldsymbol{\lambda};\boldsymbol{\rho})}[f(\boldsymbol{\lambda},\boldsymbol{\theta})] \approx E_{q(\boldsymbol{\lambda};\boldsymbol{\nu})}[f(\boldsymbol{\lambda},\boldsymbol{\rho})] \approx f(\boldsymbol{\nu},\boldsymbol{\rho})$ is that the function $f(\boldsymbol{\lambda},\boldsymbol{\theta}) = \log \frac{p(\boldsymbol{\lambda})}{q(\boldsymbol{\lambda})} + \log \frac{p(\boldsymbol{\theta})}{q(\boldsymbol{\theta}|\boldsymbol{\lambda})} + \log p(Y|\boldsymbol{\lambda},\boldsymbol{\theta},X)$ is not differentiable with respect to $D_\ell$ *when there is a jump* in the quantile evaluated at $k$, so it would seem we cannot apply it. This also happens in Nazaret & Blei (2022), Equation 11, when the function is a neural network containing non-differentiable activations like ReLU, but we could not find a discussion about it. Importantly, the function is not differentiable at a countable set of points (one for each possible width), which has a measure null. Since the Lebesgue integral of the expectation is over reals, we should be able to remove this set from the integral while ensuring that the approximation still holds mathematically.

## D    LIPSCHITZ CONTINUITY OF THE ELBO

Whenever the quantile of the finite distribution changes, we need to change the number of neurons of the neural network. Here, we want to show that the change in the ELBO after a change in the number

of neurons is bounded. Therefore, we provide a theoretical result (theorem D.1) that shows that the ELBO satisfies the Lipschitz continuity.

**Proposition D.1** (AWN ELBO Lipschitz continuity). *The ELBO loss of eq. (9), with respect to the change in the depth $D_\ell$ for the layer $\ell$, is Lipschitz continuous.*

*Proof.* We focus on the term involving $D_\ell$ of the ELBO, we write eq. (9) as

$$\log \frac{p(\boldsymbol{\nu})}{q(\boldsymbol{\nu})} + \log \frac{p(\boldsymbol{\rho})}{q(\boldsymbol{\rho}|\boldsymbol{\nu})} + \log p(Y|\boldsymbol{\nu}, \boldsymbol{\rho}, X)$$

where only the second and last terms depend on $\mathbf{D} = \{D_\ell\}_{\ell=1}^L$ . Let's first define

$$\log \frac{p(\boldsymbol{\rho})}{q(\boldsymbol{\rho}|\boldsymbol{\nu})} = \sum_{\ell=1}^L \sum_{n=1}^{D_\ell} \log \frac{p(\rho_n^\ell)}{q(\rho_n^\ell|\boldsymbol{\nu})} = \sum_{\ell=1}^L f_1(D_\ell)$$

We have that $f_1(D_\ell)$ is Lipschitz continuous, indeed, when $D_\ell$ changes to $D_{l'}$, we have

$$|f_1(D_{\ell'}) - f_1(D_\ell)| = |\sum_{n=D_\ell}^{D_{\ell'}} \log \frac{p(\boldsymbol{\rho}_n)}{q(\boldsymbol{\rho}_n|\boldsymbol{\nu})}| \tag{17}$$

$$\leq \sum_{n=D_\ell}^{D_{\ell'}} |\log \frac{p(\boldsymbol{\rho}_n)}{q(\boldsymbol{\rho}_n|\boldsymbol{\nu})}| \tag{18}$$

$$\leq \max_n |\log \frac{p(\boldsymbol{\rho}_n)}{q(\boldsymbol{\rho}_n|\boldsymbol{\nu})}||D_{\ell'} - D_\ell| \tag{19}$$

Therefore

$$|f_1(D_{\ell'}) - f_1(D_\ell)| \leq M|D_{\ell'} - D_\ell|$$

with $M = \max_n |\log \frac{p(\boldsymbol{\rho}_n)}{q(\boldsymbol{\rho}_n|\boldsymbol{\nu})}|$. We now look at the last term,

$$f_2(\mathbf{D}) = \log p(Y|\boldsymbol{\nu}, \boldsymbol{\rho}, X)$$

This function is a neural network followed by the squared norm. Therefore, $f_2(\mathbf{D})$ is continuous almost everywhere (Virmaux & Scaman, 2018), therefore Lipschitz on both parameters and the input. When we change $D_\ell$ to $D_{\ell'}$, we are adding network parameters, in particular, if we look at the two-layer network, we have

$$y = V\sigma(Wx + b) + c, V \in \mathbb{R}^{m \times n}, W \in \mathbb{R}^{n \times d}$$

we then have new parameters, which can be shown to be

$$y' = V\sigma(Wx + b) + c + \underbrace{V'\sigma(W'x + b') + c'}_{\text{new neurons}}, V' \in \mathbb{R}^{m \times n'}, W' \in \mathbb{R}^{n' \times d}, b' \in \mathbb{R}^{n'}, c' \in \mathbb{R}^m$$

the second term is also Lipschitz, therefore the function $f_2(\mathbf{D})$ is also Lipschitz. Indeed the network is composed of a linear operation and the element-wise activation functions. If the activation functions are continuous and of bounded gradient, then the whole network is Lipschitz continuous. $\square$

## E  FULL DERIVATION OF THE RESCALED WEIGHT INITIALIZATION

This Section derives the formulas for the rescaled weight initialization both in the case of ReLU and activations like tanh.

**Background**  We can rewrite Equation 10 as

$$h_i^\ell = \alpha_i^\ell p_i^\ell \tag{20}$$

$$\alpha_i^\ell = \sigma \left( \sum_{j=1}^{D_{\ell-1}} w_{ij}^\ell \underbrace{\alpha_j^{\ell-1} p_j^{\ell-1}}_{h_j^{\ell-1}} \right) \tag{21}$$

where $p_i^\ell = f_\ell(i)$.

As a refresher, the chain rule of calculus states that, given two differentiable functions $g : \mathbb{R}^D \to \mathbb{R}$ and $f = (f_1, \dots, f_D) : \mathbb{R} \to \mathbb{R}^D$, their composition $g \circ f : \mathbb{R} \to \mathbb{R}$ is differentiable and

$$(g \circ f)'(t) = \nabla g(f(t))^T f'(t)$$

$$\nabla g(f(t)) = \left( \frac{\partial g(f_1(t))}{\partial f_1(t)}, \dots, \frac{\partial g(f_D(t))}{\partial f_D(t)} \right) \in \mathbb{R}^{1 \times D}$$

$$f'(t) = (f_1'(t), \dots, f_D'(t)) \in \mathbb{R}^{D \times 1}$$

For reasons that will become clear later, we may want to compute the gradient of the loss function with respect to the intermediate activations $\boldsymbol{\alpha}^\ell$ at a given layer, that is $\nabla \mathcal{L}(\boldsymbol{\alpha}^\ell) = \left( \frac{\partial \mathcal{L}(\alpha_1^\ell)}{\partial \alpha_1^\ell}, \dots, \frac{\partial \mathcal{L}(\alpha_{N^\ell}^\ell)}{\partial \alpha_{N^\ell}^\ell} \right)$.

We focus on the $i$-th partial derivative $\frac{\partial \mathcal{L}(\alpha_i^\ell)}{\partial \alpha_i^\ell}$, where the only variable is $\alpha_i^\ell$. Then, we view the computation of the loss function starting from $\alpha_i^\ell$ as a composition of a function $\boldsymbol{\alpha}^{\ell+1} : \mathbb{R} \to \mathbb{R}^{N^{\ell+1}} = \left( \alpha_1^{\ell+1}(\alpha_i^\ell) \dots, \alpha_{N^{\ell+1}}^{\ell+1}(\alpha_i^\ell) \right)$ and another function (abusing the notation) $\mathcal{L} : \mathbb{R}^{N^{\ell+1}} \to \mathbb{R}$ that computes the loss value starting from $\boldsymbol{\alpha}^{\ell+1}$. By the chain rule:

$$\underbrace{\frac{\partial \mathcal{L}(\alpha_i^\ell)}{\partial \alpha_i^\ell}}_{(g \circ f)'(t)} = \underbrace{\left( \frac{\partial \mathcal{L}(\alpha_1^{\ell+1})}{\partial \alpha_1^{\ell+1}}, \dots, \frac{\partial \mathcal{L}(\alpha_{N^{\ell+1}}^{\ell+1})}{\partial \alpha_{N^{\ell+1}}^{\ell+1}} \right)^T}_{\nabla g(f(t))^T} \underbrace{\left( \frac{\partial \alpha_1^{\ell+1}(\alpha_i^\ell)}{\partial \alpha_i^\ell}, \dots, \frac{\partial \alpha_{N^{\ell+1}}^{\ell+1}(\alpha_i^\ell)}{\partial \alpha_i^\ell} \right)}_{f'(t)} \tag{22}$$

**Theorem 3.1** *Let us consider an MLP with activations as in Equation 21. Let us also assume that the inputs and the parameters have been sampled independently from a Gaussian distribution with zero mean and variance $\sigma^2 = \mathrm{Var}[w_{ij}^\ell] \, \forall i, j, \ell$. At initialization, the variance of the responses $\alpha_i^\ell$ across layers is constant if, $\forall \ell \in \{1, \dots, L\}$*

$$\mathrm{Var}[w^\ell] = \frac{1}{\sum_j^{D_{\ell-1}} \left( p_j^{\ell+1} \right)^2} \quad \text{for activation } \sigma \text{ such that } \sigma'(0) \approx 1 \tag{23}$$

$$\mathrm{Var}[w^\ell] = \frac{2}{\sum_j^{D_{\ell-1}} \left( p_j^{\ell+1} \right)^2} \quad \text{for the ReLU activation.} \tag{24}$$

*In addition, we provide closed form formulas to to preserve the variance of the gradient across layers.*

*Proof.* Let us start from the first case of $\sigma'(0) \approx 1$. Using the Taylor expansions for the moments of functions of random variables as in Glorot & Bengio (2010)

$$\mathrm{Var}[\alpha_i^\ell] = \mathrm{Var}\left[ \sigma \left( \sum_{j=1}^{D_{\ell-1}} w_{ij}^\ell \alpha_j^{\ell-1} p_j^{\ell-1} \right) \right] \tag{25}$$

$$\approx \sigma' \left( \mathbb{E}\left[ \sum_{j=1}^{D_{\ell-1}} w_{ij}^\ell \alpha_j^{\ell-1} p_j^{\ell-1} \right] \right) \mathrm{Var}\left[ \sum_{j=1}^{D_{\ell-1}} w_{ij}^\ell \alpha_j^{\ell-1} p_j^{\ell-1} \right] \tag{26}$$

Using the fact that $p_j$ is a constant and that $w$ and $\alpha$ are independent from each other

$$\mathbb{E}\left[ \sum_{j=1}^{D_{\ell-1}} w_{ij}^\ell \alpha_j^{\ell-1} p_j^{\ell-1} \right] = \sum_{j=1}^{D_{\ell-1}} \underbrace{\mathbb{E}[w_{ij}^\ell]}_{0} \mathbb{E}[\alpha_j^{\ell-1}] p_j^{\ell-1} = 0. \tag{27}$$

Therefore, recalling that $\sigma'(0) \approx 1$

$$\sigma' \left( \mathbb{E}\left[ \sum_{j=1}^{D_{\ell-1}} w_{ij}^\ell \alpha_j^{\ell-1} p_j^{\ell-1} \right] \right) = 1. \tag{28}$$

As a result, we arrive at

$$\mathrm{Var}[\alpha_i^\ell] \approx \mathrm{Var}\left[\sum_{j=1}^{D_{\ell-1}} w_{ij}^\ell \alpha_j^{\ell-1} p_j^{\ell-1}\right]. \tag{29}$$

Because $w$ and $\alpha$ are independent, they are also uncorrelated and their variances sum. Also, using the fact that $\mathrm{Var}[aX] = a^2\mathrm{Var}[X]$ for a constant $a$,

$$\mathrm{Var}\left[\sum_{j=1}^{D_{\ell-1}} w_{ij}^\ell \alpha_j^{\ell-1} p_j^{\ell-1}\right] = \sum_{j=1}^{D_{\ell-1}} \mathrm{Var}\left[w_{ij}^\ell \alpha_j^{\ell-1} p_j^{\ell-1}\right] = \sum_{j=1}^{D_{\ell-1}} \mathrm{Var}\left[w_{ij}^\ell \alpha_j^{\ell-1}\right]\left(p_j^{\ell-1}\right)^2 \tag{30}$$

Finally, because the mean of the independent variables involved is zero by assumption, it holds that $\mathrm{Var}[w_{ij}^\ell \alpha_j^{\ell-1}] = \mathrm{Var}[w_{ij}^\ell]\mathrm{Var}[\alpha_j^{\ell-1}]$. We can also abstract from the indexes, since the weight variables are i.i.d. and from that it follows that $\mathrm{Var}[\alpha_i^\ell] = \mathrm{Var}[\alpha_j^\ell] \ \forall i,j, \ \ i,j \in \{1,\ldots,D_\ell\}$, obtaining[3]

$$\mathrm{Var}[\alpha^\ell] \approx \mathrm{Var}[w^\ell]\mathrm{Var}[\alpha^{\ell-1}] \sum_{j=1}^{D_{\ell-1}} \left(p_j^{\ell-1}\right)^2, \tag{31}$$

noting again that the previous equation does not depend on $i$. We want to impose $\mathrm{Var}[\alpha^\ell] \approx \mathrm{Var}[\alpha^{\ell-1}]$, which can be achieved whenever

$$\mathrm{Var}[w^\ell] \approx \frac{1}{\sum_{j=1}^{D_{\ell-1}} \left(p_j^{\ell-1}\right)^2}. \tag{32}$$

**Condition on the gradients**   From a backpropagation perspective, a similar desideratum would be to ensure that $\mathrm{Var}\left[\frac{\partial\mathcal{L}(\alpha_i^\ell)}{\partial\alpha_i^\ell}\right] = \mathrm{Var}\left[\frac{\partial\mathcal{L}(\alpha_i^{\ell+1})}{\partial\alpha_i^{\ell+1}}\right]$[4].

Using Equation 22, and considering as in Glorot & Bengio (2010) that at initialization we are in a linear regime where $\sigma'(x) \approx 1$,

$$\mathrm{Var}\left[\frac{\partial\mathcal{L}(\alpha_i^\ell)}{\partial\alpha_i^\ell}\right] = \mathrm{Var}\left[\sum_{j=1}^{D_{\ell+1}} \frac{\partial\mathcal{L}(\alpha_j^{\ell+1})}{\partial\alpha_j^{\ell+1}} \frac{\partial\alpha_j^{\ell+1}(\alpha_i^\ell)}{\partial\alpha_i^\ell}\right] = \mathrm{Var}\left[\sum_{j=1}^{D_{\ell+1}} \frac{\partial\mathcal{L}(\alpha_j^{\ell+1})}{\partial\alpha_j^{\ell+1}} \frac{\partial\alpha_j^{\ell+1}(\alpha_i^\ell)}{\partial\alpha_i^\ell}\right] \tag{33}$$

$$= \mathrm{Var}\left[\sum_{j=1}^{D_{\ell+1}} \frac{\partial\mathcal{L}(\alpha_j^{\ell+1})}{\partial\alpha_j^{\ell+1}} \underbrace{\sigma'\left(\sum_{k=1}^{D_\ell} w_{jk}^{\ell+1} \alpha_k^\ell p_k^\ell\right)}_{\approx 1} w_{ji}^{\ell+1} p_i^\ell\right]. \tag{34}$$

Using the same arguments as above one can write

$$\mathrm{Var}\left[\frac{\partial\mathcal{L}(\alpha_i^\ell)}{\partial\alpha_i^\ell}\right] \approx \mathrm{Var}[w^{\ell+1}] \sum_{j=1}^{D_{\ell+1}} \mathrm{Var}\left[\frac{\partial\mathcal{L}(\alpha_j^{\ell+1})}{\partial\alpha_j^{\ell+1}} p_i^\ell\right] = \mathrm{Var}[w^{\ell+1}](p_i^\ell)^2 \sum_{j=1}^{D_{\ell+1}} \mathrm{Var}\left[\frac{\partial\mathcal{L}(\alpha_j^{\ell+1})}{\partial\alpha_j^{\ell+1}}\right]. \tag{35}$$

---

[3]It is worth noting that, in standard MLPs, $p_j^\ell = 1$ so we recover the derivation of Glorot & Bengio (2010), since $\sum_{j=1}^{D_{\ell-1}} \left(p_j^{\ell-1}\right)^2$ would be equal to $D_{\ell-1}$. In Glorot & Bengio (2010) our $D_{\ell-1}$ is denoted as "$n_{i'}$" (see Equation 5).

[4]Note that what we impose is different from Glorot & Bengio (2010), where the specific position $i$ is irrelevant. Here, we are asking that the variance of the gradients for neurons in the same position $i$, but at different layers, stays constant. Alternatively, we could impose an equivalence for all $i \neq i'$.

Expanding, we get

$$\text{Var}[w^{\ell+1}](p_i^\ell)^2 \sum_{j=1}^{D_{\ell+1}} \text{Var}\left[\frac{\partial \mathcal{L}(\alpha_j^{\ell+1})}{\partial \alpha_j^{\ell+1}}\right] \tag{36}$$

$$= (p_i^\ell)^2 \left(\prod_{i=\ell+1}^{\ell+2} \text{Var}[w^{\ell+1}]\right) \sum_{j=1}^{D_{\ell+1}} (p_j^{\ell+1})^2 \underbrace{\left(\sum_{j'=1}^{D_{\ell+2}} \text{Var}\left[\frac{\partial \mathcal{L}(\alpha_{j'}^{\ell+2})}{\partial \alpha_{j'}^{\ell+2}}\right]\right)}_{\text{constant w.r.t. } j} \tag{37}$$

$$= (p_i^\ell)^2 \left(\prod_{i=\ell+1}^{\ell+2} \text{Var}[w^{\ell+1}]\right) \left(\sum_{j'=1}^{D_{\ell+2}} \text{Var}\left[\frac{\partial \mathcal{L}(\alpha_{j'}^{\ell+2})}{\partial \alpha_{j'}^{\ell+2}}\right]\right) \left(\sum_{j=1}^{D_{\ell+1}} (p_j^{\ell+1})^2\right) \tag{38}$$

Therefore, we can recursively expand these terms and obtain

$$\text{Var}\left[\frac{\partial \mathcal{L}(\alpha_i^\ell)}{\partial \alpha_i^\ell}\right] = (p_i^\ell)^2 \left(\prod_{k=\ell+1}^{L} \text{Var}[w^k]\right) \left(\sum_{j'=1}^{D_L} \text{Var}\left[\frac{\partial \mathcal{L}(\alpha_{j'}^{D_L})}{\partial \alpha_{j'}^{D_L}}\right]\right) \left(\prod_{k=\ell+1}^{L-1} \sum_{j_k=1}^{D_k} (p_{j_k}^k)^2\right). \tag{39}$$

In this case, the variance depends on $i$ just for the term $(p_i^\ell)^2$. Finally, by imposing

$$\text{Var}\left[\frac{\partial \mathcal{L}(\alpha_i^\ell)}{\partial \alpha_i^\ell}\right] = \text{Var}\left[\frac{\partial \mathcal{L}(\alpha_i^{\ell+1})}{\partial \alpha_i^{\ell+1}}\right] \tag{40}$$

and simplifying common terms we obtain

$$(p_i^\ell)^2 \text{Var}[w^{\ell+1}] \left(\sum_{j=1}^{D_{\ell+1}} (p_j^{\ell+1})^2\right) = (p_i^{\ell+1})^2 \tag{41}$$

$$\text{Var}[w^{\ell+1}] = \frac{(p_i^{\ell+1})^2}{(p_i^\ell)^2 \sum_{j=1}^{D_{\ell+1}} \left(p_j^{\ell+1}\right)^2}. \tag{42}$$

Both Equations 32 and 42 are verified when we initialize the distributions $f_\ell$ in the same way for all layers, which implies $(p_i^\ell)^2 = (p_i^{\ell+1})^2$. Note that without this last requirement, Equation 42 would violate the i.i.d. assumption of the weights.

To show a similar initialization for ReLU activations, we need the following lemma.

**Lemma E.1.** *Consider the ReLU activation $y = \max(0, x)$ and a symmetric distribution $p(x)$ around zero. Then $\mathbb{E}[y^2] = \frac{1}{2} Var[x]$.*

*Proof.*

$$\mathbb{E}[y^2] = \int_{-\infty}^{+\infty} \max(0, x)^2 p(x) dx = \int_0^{+\infty} x^2 p(x) dx = \frac{1}{2} \int_{-\infty}^{+\infty} x^2 p(x) dx \tag{43}$$

Because $p(x)$ is symmetric, $\mathbb{E}[x] = 0$. Then

$$\frac{1}{2} \int_{-\infty}^{+\infty} x^2 p(x) dx = \frac{1}{2} \int_{-\infty}^{+\infty} (x - \mathbb{E}[x])^2 p(x) dx = \frac{1}{2} \text{Var}[x]. \tag{44}$$

$\square$

Using Lemma E.1, we can rewrite Equation 26 as

$$\text{Var}[\alpha_i^\ell] = \text{Var}\left[\sigma\left(\sum_{j=1}^{D_{\ell-1}} w_{ij}^\ell \alpha_j^{\ell-1} p_j^{\ell-1}\right)\right] = \frac{1}{2} \text{Var}\left[\sum_{j=1}^{D_{\ell-1}} w_{ij}^\ell \alpha_j^{\ell-1} p_j^{\ell-1}\right] \tag{45}$$

$$= \frac{1}{2} \text{Var}[w^\ell] \text{Var}[\alpha^{\ell-1}] \sum_{j=1}^{D_{\ell-1}} \left(p_j^{\ell-1}\right)^2 \tag{46}$$

where no approximation has been made. Similar to prior results, it follows that

$$\mathrm{Var}[w^\ell] = \frac{2}{\sum_{j=1}^{D_{\ell-1}} \left(p_j^{\ell-1}\right)^2}. \tag{47}$$

From a backpropagation perspective, since $\sigma'(x) \not\approx 1$, we use similar arguments as in He et al. (2015): we assume that the pre-activations $\alpha$ have zero mean, then the derivative of the ReLU can have values zero or one with equal probability. One can show, by expanding the definition of variance, that $\mathrm{Var}[\sigma'\left(\sum_{k=1}^{D_\ell} w_{jk}^{\ell+1}\alpha_k^\ell p_k^\ell\right) w_{ji}^{\ell+1}] = \frac{1}{2}\mathrm{Var}[w_{ji}^{\ell+1}]$. As a result, one can re-write Equation 26 as

$$\mathrm{Var}\left[\sum_{j=1}^{D_{\ell+1}} \frac{\partial \mathcal{L}(\alpha_j^{\ell+1})}{\partial \alpha_j^{\ell+1}}\sigma'\left(\sum_{k=1}^{D_\ell} w_{jk}^{\ell+1}\alpha_k^\ell p_k^\ell\right) w_{ji}^{\ell+1} p_i^\ell\right] \tag{48}$$

$$\approx \frac{1}{2}\mathrm{Var}[w^{\ell+1}](p_i^\ell)^2 \sum_{j=1}^{D_{\ell+1}} \mathrm{Var}\left[\frac{\partial \mathcal{L}(\alpha_j^{\ell+1})}{\partial \alpha_j^{\ell+1}}\right], \tag{49}$$

where we have made the assumption of independence between $\sigma'\left(\sum_{k=1}^{D_\ell} w_{jk}^{\ell+1}\alpha_k^\ell p_k^\ell\right)$ and $w_{ji}^{\ell+1}$ since the former term is likely a constant. Following an identical derivation as above, we obtain

$$\mathrm{Var}[w^{\ell+1}] = \frac{2(p_i^{\ell+1})^2}{(p_i^\ell)^2 \sum_{j=1}^{D_{\ell+1}} \left(p_j^{\ell+1}\right)^2} \tag{50}$$

which is almost identical to Equation 47 and not dependent on $i$ when we initialize $f_\ell$ in the same way for all layers. □

**Comparison of convergence for different initializations** Below, we provide a comparison of convergence between an AWN ReLU MLP initialized with the standard Kaiming scheme and one initialized according to our theoretical results ("Kaiming+").

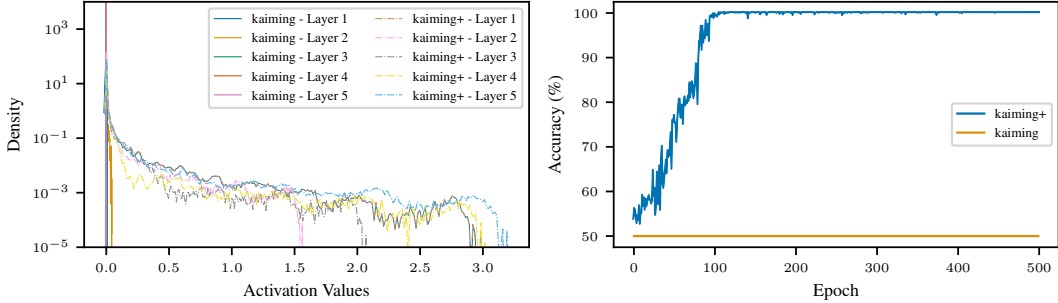

Figure 7: (Left) Effect of the rescaled initialization scheme ("Kaiming+") on a ReLU-based MLP, where neurons' activations are computed using Equation 10. Compared to the standard initialization, the variance of activations agrees with the theoretical result. (Right) Without the rescaled initialization, convergence is hard to attain on the SpiralHard dataset (please refer to the next section for details about the dataset).

## F  DATASET INFO AND STATISTICS

The synthetic datasets DoubleMoon, Spiral, and SpiralHard are shown in Figure 8. These binary classification datasets have been specifically created to analyze the behavior of AWN in a controlled scenario where we are sure that the difficulty of the task increases. We provide the code to generate these datasets in the supplementary material.

Table 3 reports information on all datasets' statistics and the data splits created to carry out model selection (inner split) and risk assessment (outer split), respectively. Note that for Multi30k, the

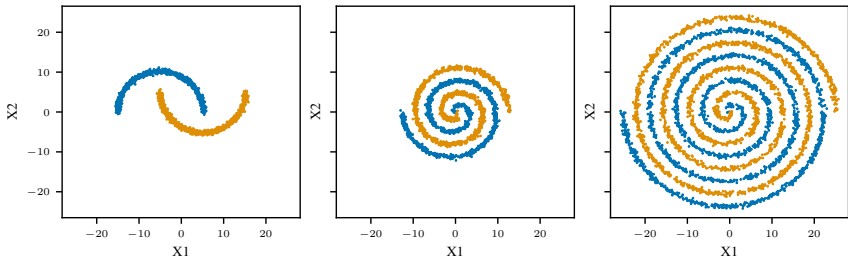

Figure 8: The DoubleMoon, Spiral, and SpiralHard synthetic datasets are used to test AWN's inner workings, and are ordered difficulty. Orange and blue colors denote different classes.

Table 3: Dataset statistics and number of samples in each split are shown.

|  | # Samples | # Features | # Classes | Outer Split (TR/VL/TE) | Inner Split (TR/VL) |
|---|---|---|---|---|---|
| DoubleMoon | 5000 | 2 | 2 | 3600/400/1000 | 3600/400 |
| Spiral | 5000 | 2 | 2 | 3600/400/1000 | 3600/400 |
| SpiralHard | 10000 | 2 | 2 | 7200/800/2000 | 7200/800 |
| pol | 15000 | 48 | 2 | 10800/1200/3000 | 10800/1200 |
| MiniBooNE | 130064 | 50 | 2 | 93645/10406/26013 | 93645/10406 |
| credit card clients | 30000 | 23 | 2 | 21600/2400/6000 | 21600/2400 |
| MNIST | 70000 | 28x28 | 10 | 48610/9723/11667 | 48610/9723 |
| CIFAR10 | 60000 | 32x32x3 | 10 | 41665/8334/10001 | 41665/8334 |
| CIFAR100 | 60000 | 32x32x3 | 100 | 41665/8334/10001 | 41665/8334 |
| NCI1 | 4110 | 37 | 2 | 3328/370/412 | 3328/370 |
| REDDIT-B | 2000 | 1 | 2 | 1620/180/200 | 1620/180 |
| Multi30k | 31000 | 128x50257 | Multilabel | 29046/968/1000 | 29046/968 |

maximum sequence length is 128 and the number of possible tokens is 50527. The classification loss is computed by comparing the predicted token against the expected one, ignoring any token beyond the length of the actual target. For a detailed description of the graph datasets, the reader can refer to Errica et al. (2020).

## G  HYPER-PARAMETERS FOR FIXED AND AWN MODELS

For each data domain, we discuss the set of hyper-parameters tried during model selection for the fixed baseline and for AWN. We try our best to keep architectural choices identical with the exception of the width of the MLPs used by each model. In all experiments, we either run patience-based early stopping (Prechelt, 1998) or simply select the epoch with best validation score.

### G.1  SYNTHETIC TABULAR DATASETS AND PERMUTED MNIST

We report the hyper-parameter configurations tried for the fixed MLP/RNN and AWN in Table 4. In particular, the starting width of AWN under the quantile function evaluated at $k = 0.9$ is approximately 256 hidden neurons. We also noticed that a learning rate value of $0.1$ makes learning the width unstable, so we did not use it in our experiments. The total number of configurations for the fixed models is 180, for AWN is 18. We also did not impose any prior on an expected size to see if AWN recovers a similar width compared to the best fixed model (please refer to Table 1) starting from the largest configuration among the fixed models.

### G.2  IMAGE CLASSIFICATION DATASETS

We train from scratch a ResNet20 and focus our architectural choices on the MLP that performs classification using the flattened representation provided by the previous CNN layers. This ensures that any change is performance is only due to the changes we apply to the MLP. The configurations

Table 4: Hyper-parameter configurations for standard MLP/RNN and AWN versions on tabular datasets.

| Hyper-Parameter | DoubleMoon | Spiral | SpiralHard/pol/MiniBooNE/credit c. |
|---|---|---|---|
| Batch Size | 32 | [32, 128] (MLP/RNN), 128 (AWN) | [32, 128] (MLP/RNN), 128 (AWN) |
| Epochs | 500 | 1000 | 5000 |
| Hidden Layers | 1 | 1 | [1, 2, 4] |
| Layer Width | | [8, 16, 24, 128, 256] | |
| Non-linearity | | [ReLU, LeakyReLU, ReLU6] | |
| Optimizer | | Adam, learning rate $\in$ [0.1, 0.01] (MLP/RNN), 0.01 (AWN) | |
| **AWN Specific** | | | |
| Quantile Fun. Threshold $k$ | | 0.9 | |
| Exponential Distributions Rate | | 0.01 | |
| $\sigma_\ell^\theta$ | | [1.0, 10.0] | |
| Prior over $\lambda$ | | Uninformative | |

are shown in Table 5; note that a width of 0 implies a linear classifier instead of a 1 hidden layer MLP. After previous results on tabular datasets showed that a LeakyReLU performs very well, we fixed it in the rest of the experiments. In this case, we test 5 different configurations of the fixed baseline,

Table 5: Hyper-parameter configurations for standard MLP and AWN versions on image classification datasets.

| Hyper-Parameter | MNIST / CIFAR10 / CIFAR100 |
|---|---|
| Batch Size | 128 |
| Epochs | 200 |
| Layer Width (classification layer) | [0, 32, 128, 256, 512] |
| Optimizer | SGD, learning rate 0.1, weight decay 0.0001, momentum 0.9 |
| Scheduler | Multiply learning rate by 0.1 at epochs 50, 100, 150 |
| **AWN Specific** | |
| Quantile Fun. Threshold $k$ | 0.9 |
| Exponential Distributions Rate | 0.02 |
| $\sigma_\ell^\theta$ | 1.0 |
| Prior over $\lambda$ | Uninformative |

whereas we do not have to perform any model selection for AWN. The rate of the exponential distribution has been chosen to have a starting width of approximately 128 neurons, which is the median value among the ones tried for the fixed model.

## G.3 TEXT TRANSLATION TASKS

Due to the cost of training a Transformer architecture from scratch on a medium size dataset like Multi30k, we use most hyper-parameters' configurations of the base Transformer in Vaswani et al. (2017), with the difference that AWN is applied to the MLPs in each encoder and decoder layer. We train an architecture with 6 encoder and 6 decoder layers for 500 epochs, with a patience of 250 epochs applied to the validation loss. We use an Adam optimizer with learning rate 0.01, weight decay 5e-4, and epsilon 5e-9. We also introduce a scheduler that reduces the learning rate by a factor 0.9 when a plateau in the loss is reached. The number of attention heads is set to 8 and the dropout to 0.1. We try different widths for the MLPs, namely [128,256,512,1024,2048]. The embedding size is fixed to 512 and the batch size is 128. The AWN version starts with an exponential distribution rate of 0.004, corresponding to approximately 512 neurons, and a $\sigma_\ell^\alpha$=1.

## G.4 GRAPH CLASSIFICATION TASKS

We train the AWN version of Graph Isomorphism Network, following the setup of Errica et al. (2020) and reusing the published results. We test 12 configurations as shown in the table below. Please note that an extra MLP with 1 hidden layer is always placed before the sequence of graph convolutional layers.

Table 6: Hyper-parameter configurations for AWN versions on graph classification datasets.

| Hyper-Parameter | NCI1 / REDDIT-B |
|---|---|
| Batch Size | [32,128] |
| Epochs | 1000 |
| Graph Conv. Layers | [1, 2, 4] |
| Global Pooling | [sum, mean] |
| Optimizer | Adam, learning rate 0.01 |
| **AWN Specific** | |
| Patience | 500 epochs |
| Quantile Fun. Threshold $k$ | 0.9 |
| Exponential Distributions Rate | 0.02 |
| $\sigma_\ell^\theta$ | 10.0 |
| Prior over $\lambda$ | Uninformative |

## H    RE-TRAINING A FIXED NETWORK WITH THE LEARNED WIDTH

While on DoubleMoon AWN learns a similar number of neurons chosen by the model selection procedure and the convergence to the perfect solution over 10 runs seem much more stable, this is not the case on Spiral, SpiralHard and REDDIT-B, where AWN learns a higher total width over layers. To investigate whether the width learned by AWN on these datasets is indeed the reason behind the better performances (either a more stable training or better accuracy), we run again the experiments on the baseline networks, this time by fixing the width of each layer to the average width over hidden layers learned by AWN and reported in Table 1. We call this experiment "Fixed+".

| | Spiral | SpiralHard | REDDIT-B |
|---|---|---|---|
| Fixed | 99.5(0.5) | 98.0(2.0) | 87.0(4.4) |
| AWN | 99.8(0.1) | 100.0(0.0) | 90.2(1.3) |
| Fixed+ | 100.0(0.1) | 83.2(17.0) | 90.7(1.4) |

The results suggest that by using more neurons than those selected by grid search – possibly because the validation results were identical between different configurations – it is actually possible to improve both accuracy and stability of results on Spiral and REDDIT-B, while we had a convergence issue on SpiralHard. Notice that the number of neurons learned by AWN on REDDIT-B was well outside of the range investigated by Xu et al. (2019) and Errica et al. (2020), demonstrating that the practitioner's bias can play an important role on the final performance.

Because we are not aware of the best width achieved by the grid search approach of Errica et al. (2020) on REDDIT-B, we ran an additional experiment on fixed models, testing smaller and larger widths compared to the one selected by AWN (average of $\sim$793). What we discover is the following.

Table 7: We report REDDIT-B *validation* performances, averaged across the 10 outer folds, for a fixed-width DGN. The other hyper-parameters match those selected by AWN during model selection, as per this specific experiment.

| # Total Width | Mean | Std |
|---|---|---|
| 50 | 89.28 | 3.50 |
| 250 | 92.89 | 1.45 |
| 500 | 92.67 | 2.03 |
| 793 | 92.83 | 3.55 |
| 1000 | 93.06 | 1.60 |
| 1250 | 92.78 | 2.33 |
| 1500 | 91.78 | 2.41 |

Though these results are statistically comparable, the average performance obtained using AWN's learned width seems to be close to a local optimum for the fixed-width model.

## I  FURTHER QUALITATIVE RESULTS ON IMAGE CLASSIFICATION DATASETS

Because it is well known that MNIST, CIFAR10, and CIFAR100 are datasets of increasing difficulty, we attach a figure depicting the learned number of neurons and the convergence of AWN on these datasets, following the practices highlighted in He et al. (2016) and described in Table 5 to bring the ResNet20 to convergence. Akin to the tabular datasets, we see a clear pattern where the learned number of neurons, averaged over the 10 final training runs, increases together with the task difficulty. The AWN version of the ResNet20 converges in the same amount of epochs as the fixed one.

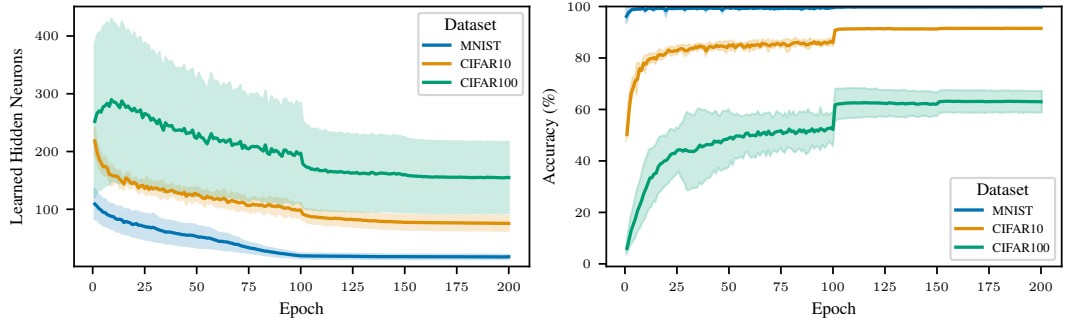

Figure 9: We present an analysis similar to Figure 2 but for image classification datasets of increasing difficulty. One can observe the jumps caused by the learning rate's scheduling strategy.

## J  COMPARISON WITH NEURAL ARCHITECTURE SEARCH (NAS)

Although complementary to our approach, we thought instructive to run a some basic NAS methods on the image classification datasets. Most NAS methods fix an upper bound on the maximum width, whereas AWN's methodology does not. The truncation ability of AWN is not intended as a substitute for pruning but rather as a confirmation of our claims. In fact, NAS and pruning can actually be combined with AWN and do not necessarily be seen as "competing" strategies.

The results are shown in Table 8. That said, we compared the performance of AWN against simple NAS methods on the vision datasets. We tested grid search, random search (Following Wu et al. (2020)), local search (White et al., 2021), and Bayesian Optimization (Barber, 2012), fixing the same budget of 5 for all methods and width values between 0 (Linear classifier) and 512 neurons. Results are comparable to those of AWN.

Table 8: Comparison between different NAS approaches and AWN. We remind the reader that NAS is an orthogonal and complementary research direction to AWN.

|          | Grid Search | Random Search | Local Search | Bayes. Optim. | AWN        |
|----------|-------------|---------------|--------------|---------------|------------|
| MNIST    | 99.6(0.1)   | 99.6(0.0)     | 99.5         | 99.4          | 99.7(0.0)  |
| CIFAR10  | 91.4(0.2)   | 90.1(0.5)     | 90.6         | 91.2          | 91.4(0.2)  |
| CIFAR100 | 66.5(0.4)   | 64.9(1.1)     | 64.9         | 65.9          | 63.1(4.0)  |

## K    NON-LINEARITY ABLATION STUDY

Below, we report an ablation study on the impact of non-linear activation functions. We analyze the convergence to the same width across different batch sizes and exponential starting rates $\lambda$ on the SpiralHard dataset.

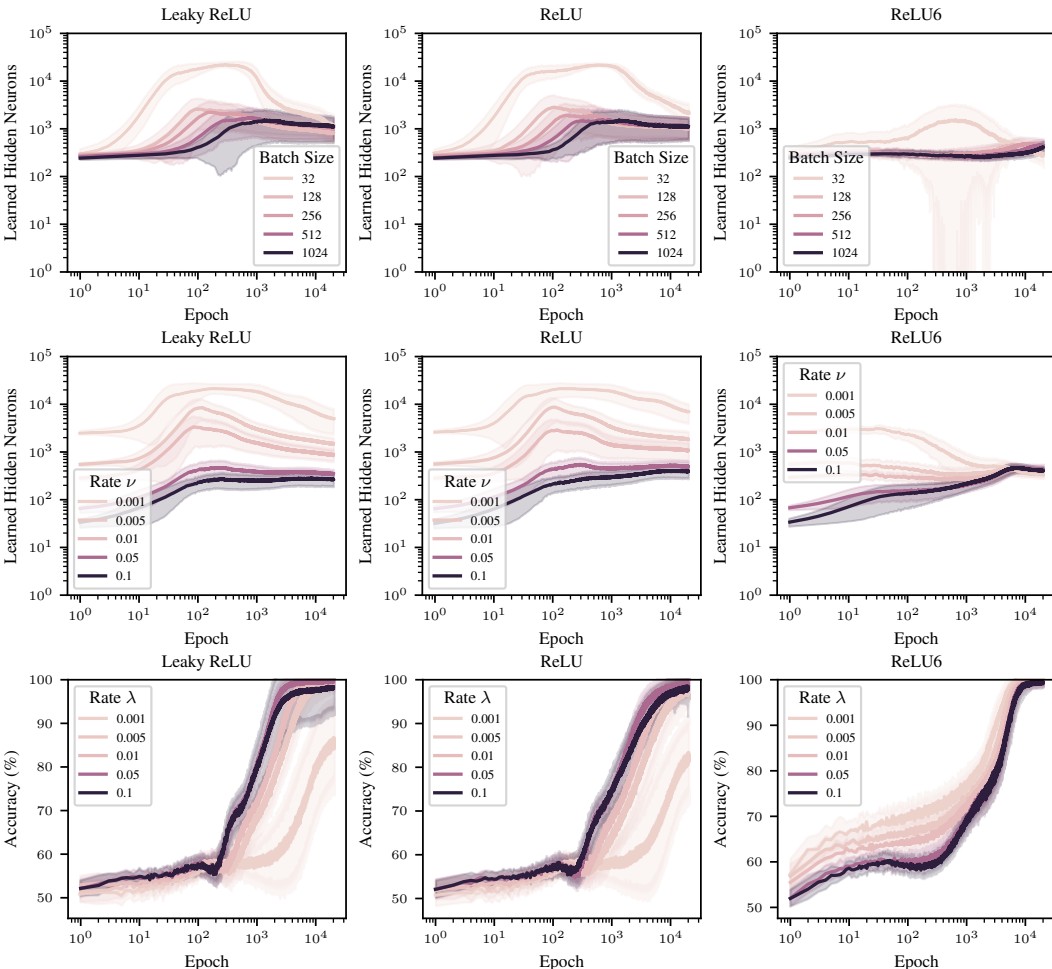

Figure 10: Impact of different batch sizes and starting exponential rates $\lambda$, organized by type of non-linear activation function.

These qualitative analyses support our claims that using a bounded activation is one way to encourage the network not to counterbalance the learned rescaling of each neuron. In fact, using a ReLU6 shows a distinctive convergence to the same amount of neurons among all configurations tried. The decreasing trend for LeakyReLU and ReLU activations may suggest that these configurations are also converging to a similar value than ReLU6, but they are taking a much longer time.

# L   LEARNED WIDTH FOR EACH LAYER ON GRAPH DATASETS

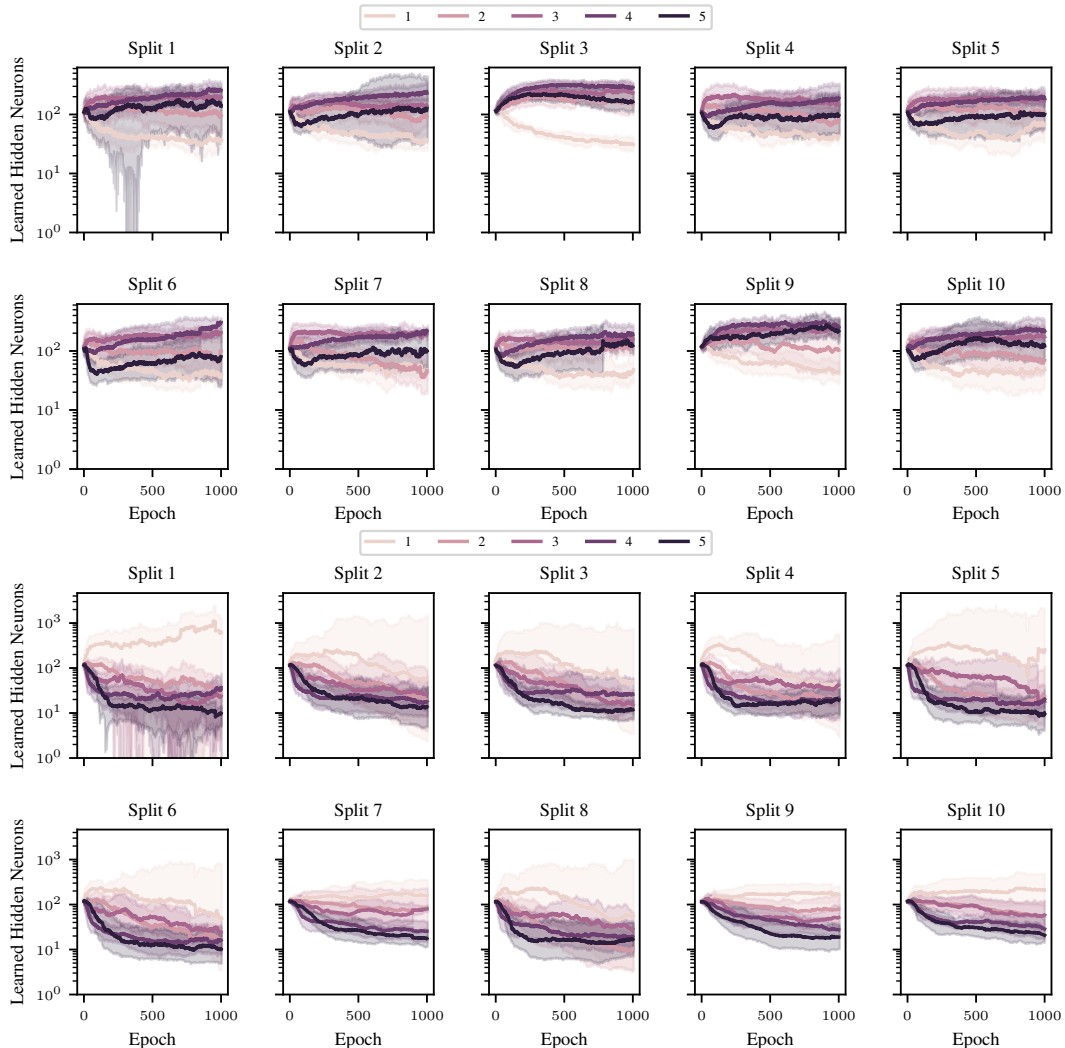

Figure 11: Learned neurons per layer (from 1 to 5), averaged over 10 final runs for each of the 10 best configurations selected in the outer folds. The first and second rows refer to NCI1, the third and fourth refer to REDDIT-B. One can observe similar trends in most cases.

# M  QUANTILE ABLATION STUDY

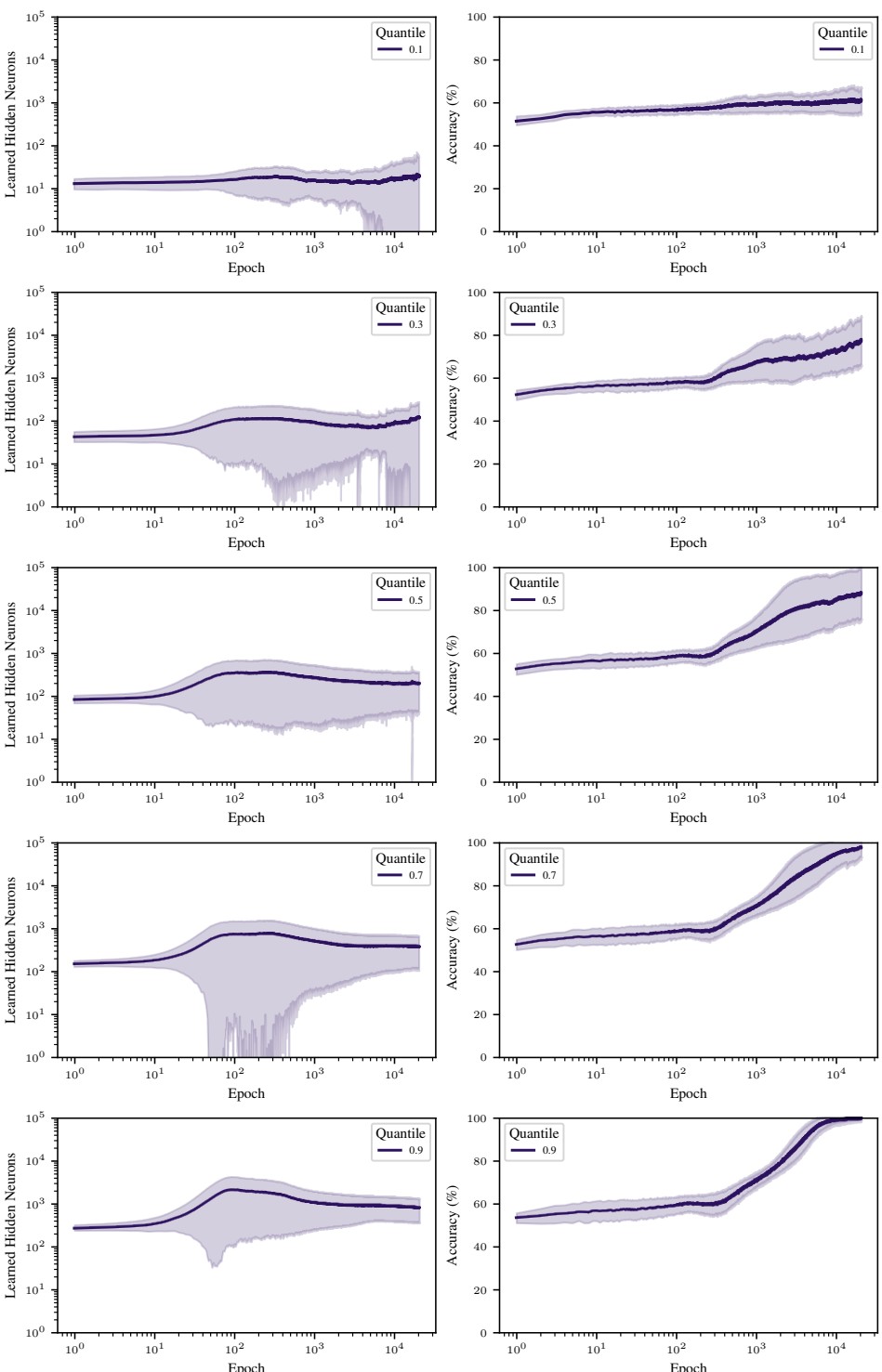

Figure 12: We report learned width and validation accuracy trends averaged over 1-layer AWN configurations on SpiralHard. We see how a higher quantile $k$ (i.e., a better ELBO approximation) grants more stable performances. All architectures keep adapting to the task.

# N    REGULARIZATION ABLATION STUDY

In Section 3, we argued there are two ways we can maximize the importance rescaling effect, namely by using bounded activation functions and by regularizing the weights. In this section, we show with an ablation that these strategies are generally not required, in the sense that AWN is still able to solve the tasks, although it might introduce a more neurons than needed. The table below compares our original results on tabular data (Table 1) with those where we did not consider ReLU6 and where we imposed an almost null regularizer on the weights ($\sigma_\ell^\theta = 100$).

Table 9: We compare performances and learned width of AWN, where we removed regularization and bounded activations from hyper-parameters, with the results in the main paper.

| | AWN | | AWN - NoReg | | Width (AWN) | | Width (AWN - NoReg) | |
|---|---|---|---|---|---|---|---|---|
| | Mean | (Std) | Mean | (Std) | Mean | (Std) | Mean | (Std) |
| DoubleMoon | 100.0 | (0.0) | 100.0 | (0.2) | 8 | (2.8) | 17 | (16.6) |
| Spiral | 99.8 | (0.1) | 100.0 | (0.0) | 66 | (8.7) | 108 | (26.8) |
| SpiralHard | 100.0 | (0.0) | 100.0 | (0.0) | 227 | (32.4) | 539 | (809.8) |
| pol | 99.2 | (0.1) | 99.2 | (0.1) | 84 | (11.0) | 2335 | (552.8) |
| MiniBooNE | 93.2 | (0.1) | 93.8 | (0.2) | 53 | (11.1) | 4907 | (1141) |
| credit card | 81.8 | (0.1) | 81.8 | (0.1) | 51 | (12.0) | 53 | (13) |

We see that in all cases, not using regularization nor bounded activations leads to comparable performances but definitely higher widths. For some of the configurations tried, the width tends to grow a lot, which is why we set a maximum width of 5000 in the interest of time, although performances do not seem to decrease in spite of that. Further investigation into the results revealed that the weight regularization is what helps the most in controlling a stable behavior of the width, while the bounded activation did not have a clear impact on that. We therefore recommend some degree of regularization to make sure that the effect of the importance rescaling is properly taken into account by the optimization process.

## O  WALL-CLOCK TIME COMPARISON

To further highlight the advantages brought by AWN, we report the average runtime (in seconds) required to complete a single model selection configuration for AWN and for the fixed model. In cases where model selection is not performed (e.g., AWN on vision tasks), we instead report the average runtime of a final run. Results for NCI1 and REDDIT are excluded from this comparison, since they were taken directly from the literature. Based on these runtimes, we compute two measures: *i)* the ratio of average single-run times (Single-Run Ratio), and *ii)* the ratio of total time required by AWN and the fixed models to complete the entire model selection process (Model Selection Ratio).

Table 10: We report average wall-clock times for AWN and the fixed model, measured either over model selection configurations or final runs when no selection is performed. We also provide two ratios: *i)* single-run wall-clock times and *ii)* total model selection time (fixed models always test at least five widths).

| Dataset | Avg Fixed (s) | Avg AWN (s) | Single-Run Ratio (↓) | Model Selection Ratio (↓) |
|---|---|---|---|---|
| DoubleMoon | 70 | 218 | 3.1× | **0.62×** |
| Spiral | 120 | 141 | 1.1× | **0.24×** |
| SpiralHard | 1982 | 7772 | 3.9× | **0.78×** |
| pol | 1251 | 1088 | **0.87×** | **0.17×** |
| MiniBooNE | 11794 | 4496 | **0.38×** | **0.07×** |
| creditcards | 2104 | 1815 | **0.90×** | **0.14×** |
| PMNIST | 842 | 2014 | 2.4× | **0.48×** |
| MNIST | 1838 | 3000 | 1.6× | **0.33×** |
| CIFAR10 | 1444 | 2880 | 2× | **0.40×** |
| CIFAR100 | 1474 | 1740 | 1.2× | **0.24×** |
| Multi30k | 45100 | 19800 | **0.42×** | **0.09×** |

The results confirm that AWN's overhead of adapting the network vanishes for larger datasets with higher feature dimensionality. Interestingly, on some of these datasets the average single-run wall time is even shorter than the fixed models. By being able to reduce hyper-parameter tuning of the width while preserving most performances, the total model selection time ratios are consistently in favor of AWN, and in principle scale favorably with the number of widths tested. Note that memory consumption scales with the width of the layers, but our methodology does not introduce any noticeable extra memory requirements.

## P    IMPACT OF REMOVING MOST IMPORTANT NEURONS

We complement the truncation analysis of Figure 5 by showing what happens when we remove individual neurons or a group of them, starting from the most important ones. This analysis provides further insights about how neurons behave in AWN.

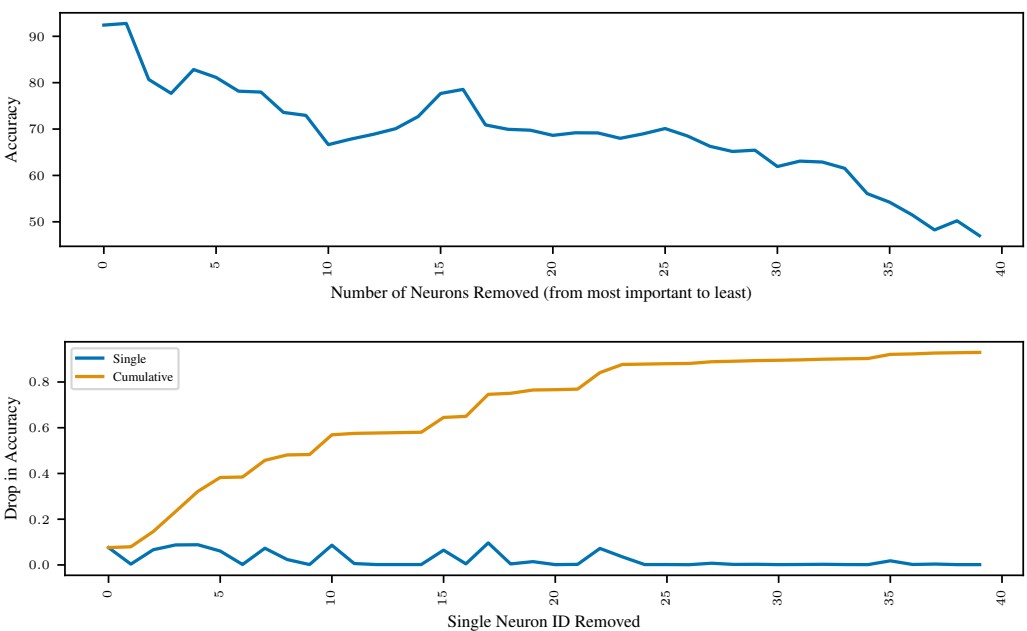

Figure 13: We show the impact of removing the most important neurons, either in batch (top) or individually (bottom) on Spiral performances, using the same experiment as Figure 5.

We observe that performance quickly drops, as expected, and we reach random performance after removing the first ∼40 neurons out of 83. The second analysis prunes individual neurons to understand how much their absence influences performance – which can have a different effect compared to removal of a set of neurons – to see if there exist two neighboring neurons that contribute similarly to the performance drop or do not contribute at all. Removing most of the first 25 neurons out of 83 contributes majorly to performance degradation, but interestingly the removal of some neurons does not affect performances. Instead, removing any subsequent neuron has little impact (but for one) on performances, as one would expect. This suggests that we may be able to compress the neural network even more: we leave this investigation to future work.

