# OpenReview forum: "Adaptive Width Neural Networks"
_ICLR.cc/2026/Conference — ICLR 2026 Poster_

### Official Review · Reviewer_A2U1 · 2025-10-28

**Soundness:** 3
**Presentation:** 3
**Contribution:** 3
**Rating:** 6
**Confidence:** 3

**Summary:**

The paper introduces Adaptive Width Neural Networks (AWNN), a probabilistic framework that maximizes a simple variational objective via backpropagation over a neural network’s parameters. Further a soft importance ordering is imposed on the neurons of the layer, whereby runtime dynamic truncation adaptively controls the width, and online compression can be realized naturally through regularization.

**Strengths:**

The paper introduces a new approach for learning netwroks of unbounded width. The paper provides a theoretical derivation for their model as well as expermients that support their claims.

**Weaknesses:**

The proposed method shares notable similarities with prior work on learning infinite depth, which may limit the perceived novelty. And there has already been some exploration in the direction of learning model's width.

**Questions:**

no questions

---

> ### Author Response · Authors · 2025-11-17
> **Response to Reviewer A2U1**
>
> We want to thank the Reviewer for the positive assessment of our work. Please allow us to comment on the novelty aspect the Reviewer identified.
>
> While it is true that we were inspired by the UDN work of Nazaret and Blei (2022), the methodology introduces several technical differences that are worth highlighting. First, learning the width requires a different formulation of the ELBO, since $\lambda$ is a continuous variable rather than the discrete truncation parameter used in UDN, which necessitated a different treatment by passing it directly into the model. Second, the family of truncated distributions must be monotonically decreasing, a requirement that was not considered in UDN because that work focused on adaptive depth, where such a bias is not appropriate. We view these as significant and nontrivial methodological distinctions that set our approach apart from UDN. Finally, as noted in the related work section, we acknowledge prior efforts on determining neural network width, but we believe our method is novel and benefits from a level of implementation simplicity that other approaches typically lack.
>
> Thank you for giving us the opportunity to clarify these points. We have revised the related work section to present these aspects more clearly.
>
> ----
>
> **Conclusions:** Finally, we would greatly appreciate it if the Reviewer could consider updating the score in light of our clarifications, as there do not appear to be major remaining concerns. We would, of course, be happy to provide any additional information the Reviewer may find helpful.

---

### Official Review · Reviewer_SBvZ · 2025-10-31

**Soundness:** 2
**Presentation:** 3
**Contribution:** 2
**Rating:** 4
**Confidence:** 2

**Summary:**

This paper presents a framework that enables neural networks to dynamically learn and adjust their layer widths during training, removing the need for preset upper bounds.
The method leverages a probabilistic variational inference approach, adapted from “Variational Inference for Infinitely Deep Neural Networks” by Nazaret and Blei (2022), but focuses on varying layer widths rather than depths.
By automating the process of width selection, the approach streamlines model design and complements traditional hyperparameter search.
Extensive experiments on both synthetic and real-world datasets demonstrate that Adaptive Width Neural Networks (AWNN) consistently match or outperform fixed-width baselines, allow efficient post-training compression via truncation, and remain robust across various hyperparameter choices and batch sizes.

**Strengths:**

1. Strong Motivation and Practical Impact:

The authors address the challenge of hyperparameter tuning, a particularly arduous and resource-intensive task for large-scale networks. Automating the search for the width hyperparameter is therefore a highly practical contribution, especially in modern architectures with billions of parameters.

2. Efficient Post-Training Trade-Off:

The proposed method offers the compelling ability to control model complexity after training with zero additional cost. This property is particularly valuable for practitioners who need to deploy models with varying computational or memory constraints, as it facilitates rapid reduction in training costs across architectures of different depths and widths.

3. Theoretical foundation:

While I am not fully familiar with variational inference frameworks and cannot fully verify the technical details of the problem formulation in Section 3, the method closely follows the established framework of Nazaret and Beli (2022), and the overall design seems to have no evident problem, but I hope that other reviewers with stronger expertise in this area will thoroughly assess the technical soundness of the approach.

**Weaknesses:**

1. Limited Demonstration of Practicality and Scalability:

Although the paper emphasizes practical implications, the experimental results do not fully demonstrate the method’s practical utility or scalability. Specifically, the approach is only applied to multi-layer perceptron (MLP) layers across various models; for convolutional neural networks (CNNs), it is restricted to the final MLP classifier layer. Given the paper’s claim of general applicability, it is unclear why the adaptive width mechanism has not been validated on other layer types, such as convolutional layers. Notably, there are several supernet approaches for adaptive width in CNNs such as [1][2], which would make for a valuable comparison to further highlight the novelty or limitations of this method.

[1] Slimmable neural networks (ICLR 2019)

[2] Alphanet: Improved training of supernets with alpha-divergence (ICML 2021)

2. Scalability and Interdependence Across Layers:

While the framework follows that of “unbounded depth neural networks” (UDN)", the task of selecting layer widths in AWNN appears even more challenging, since truncation points must be chosen for every layer rather than just the overall depth. The paper assumes interdependence across layers but does not sufficiently address this complexity. In practice, changing the width of one layer can have significant effects on subsequent layers, and the impact of these dependencies deserves more thorough exploration and discussion.

**Questions:**

1. My understanding is that $D_l$ (the truncated width of layer l) can fluctuate, increasing or decreasing, during training. For exmaple, DoubleMoon in Figure 2. In such cases, are the weights for newly added neurons initialized from scratch each time, and are discarded weights permanently removed? Alternatively, if the width decreases and then increases again later, are previously discarded neurons and their weights reinstated, or are they reinitialized?
I am concerned that frequent initialization and discarding of neuron weights could potentially slow down the optimization process.

2. The training procedure described in Algorithm 1 appears computationally efficient. Given this, would it be practical to evaluate the method on larger datasets (e.g., ImageNet) and deeper models (e.g., ResNet)? What is the actual training cost of the proposed approach in terms of runtime and memory usage? Are there any hidden or unexpected costs that might limit scalability to large datasets and architectures?

3. Regarding Table 1: does “Linear" refer to a linear classifier with the width equal to the number of dataset classes (e.g., width 10 for MNIST and CIFAR-10)? If so, does AWNN tend to learn significantly larger widths than this baseline for these tasks?

---

> ### Author Response · Authors · 2025-11-17
> **Response to Reviewer SBvZ: Part 1**
>
> We want to thank the Reviewer for highlighting the strengths of our paper, especially as regards to motivation, potential impact, and theoretical foundations. This is a really important acknowledgement for us.
>
> The reviewer mentions two weaknesses of our work, which we would like to comment on below.
>
> **Weakness 1:** We would first like to clarify that the statement of *general applicability* refers to the fact that our technique can be applied in any context where an MLP is used. We acknowledge that this was not made sufficiently explicit in the paper and will make this clearer in the revised paper, for instance in the introduction and in section 3.3.
>
> That said, our approach is indeed applicable to all families of networks that rely on MLPs within their framework — including certain CNNs, GNNs, Transformers, and RNNs — and we have empirically demonstrated this in our experiments.
>
> Regarding the reviewer’s suggestion to extend the technique to also learn the number of filters in CNNs, we would like to emphasize that this direction is explicitly identified as future work in Section 3.3. As such, it represents a natural avenue for extending our current contribution rather than a limitation of it. Since this extension requires a distinct methodological formulation, it lies beyond the scope of the present work and therefore cannot reasonably be considered a weakness of this manuscript.
>
> We want to thank the Reviewer for the references on CNN supernets: we will add them in the Neural Architecture Search background section and consider them as baselines for future extensions of our methodology to CNNs filters.
>
> **Weakness 2:** We are not sure what the Reviewer refers to when mentioning scalability here, since there is no reference to it in the text below the bullet point. We will therefore comment on the interdependence between layers.
>
> We agree that subsequent layers can influence each other, however, the layers’ size changes are always associated with neurons with little importance, which is why training is quite stable in all our experiments. A neuron with a tiny average activation will not contribute much to the activations of subsequent neurons, therefore, there is generally no cause for concern.
>
> ----
> We will now try to provide clarifications to the question posed by the Reviewer.
>
> **Question 1:** We opted to reinitialize the weights from scratch due to the simplicity of this approach. Empirically, we did not observe noticeably slower convergence. This is because newly introduced neurons have relatively small importance compared to existing ones, and the width increases primarily when previously present neurons require higher importance according to the gradient. As a result, fluctuations in the tail should have limited impact. This observation is related to the previous point, Weakness 2.
>
> To clarify more, what the Reviewer observes in Figure 2 (left) for DoubleMoon does not reflect instability of a single run, but rather the average over the final training runs that are all converging to similar values. However, since the y-axis is in log scale the behavior appears more fluctuations when the learned width is between 0 and 10. Indeed, the model has already converged to a 100% accuracy when this behavior occurs, and it is due to models trying to further compress their width.
>
>
> **Question 2:** The training procedure is indeed computational efficient, as the overhead to update the layers width is minimal compared to the forward pass of the neural network. It is definitely practical to apply this to deeper networks, for instance the Transformer architecture in the Multi30k experiment. Regarding ResNets, we note that extending the methodology to CNNs is an interesting direction for future work, but it is beyond the scope of the present paper.
>
>
> In Appendix E, we also derived theoretical results about our rescaled Kaiming initialization that guarantee that the gradient flows as in standard deep ReLU networks (see Figure 7 for a concrete visualization), so we do not foresee any practical issue in applying AWNN to deep networks. In terms of training costs, please refer to Appendix O, where we compute the average cost of single AWN runs vs fixed model as well as the cost to perform hyper-parameter tuning. Note that training costs scale with the width learned by AWNN during training, so in some cases it is higher for a single run, but AWNN avoids the need to properly cross-validate the width of neural networks, which positively affect the total running costs for model selection. Finally, memory consumption scales with the width of the layers, but our methodology does not introduce any noticeable extra memory requirements, which is why we did not make any considerations before. We have revised the paper to make this clearer

---

> > ### Author Response · Authors · 2025-11-17
> > **Response to Reviewer SBvZ: Part 2**
> >
> > **Question 3:** “Linear” refers to a linear classifier, with no hidden layers. Hence we cannot compare the number of classes of the linear layer with the width of the MLP’s hidden layer, but we do not consider the width learned by AWN for MNIST and CIFAR10 “significantly large”, as 20 and 80 neurons are typically a small amount. In practice, one can choose between a linear classifier and an adaptive-width MLP during model selection, so that the linear can be selected in case there is no need for a more complicated architecture.
> >
> > ----
> >
> > **Conclusions:** We hope these clarifications helped to better explain the scope of our work and to address the technical concerns raised by the Reviewer. We would greatly appreciate it if the Reviewer could consider updating the score in light of our responses. We remain available for any further clarification.

---

### Official Review · Reviewer_KoUG · 2025-10-31

**Soundness:** 3
**Presentation:** 3
**Contribution:** 3
**Rating:** 8
**Confidence:** 4

**Summary:**

The paper investigates growing the width of the linear layers, by considering the width to be hypothetically infinite dimensional, they then estimate the number of neurons in a layer during each batch by a single learnt latent variable $v_l$ per layer, which shifts the importance distribution (they have a fixed monotonically decreasing function to compute the probability, that is the initial neurons are more likely to be important than the new ones), any new neurons within the cumulative probability threshold ( $k$ >0.9)  are added, and the neurons with the least index (importance) can be de. Since new neurons are expected to have low importance (as the function is monotonically decreasing) the addition is expected to be stable. The experimentation is performed on tablular, image and text data on MLP, CNN and Text respectively to show adaptability of the method.

**Strengths:**

1. The main advantage of the paper is that the latent variable which controls the width is learnt, so the addition and deletion of neurons is fast and requires significantly less compute than existing hessian based methods.
2. The training is stable at appropriate batch sizes, since the importance is low for large widths, and usually we start with a considerably large width for large models.
3. The strict ordering based on index ensures that old neurons are preserved, which again improves the stability of the training, but at the cost of flexibility.
4. The results show that usually, the training yields similar or better accuracy, with similar width and in some cases, the width is dramatically increased but the performance gains too.

**Weaknesses:**

1. Its harder for the neural network to change trajectory, since the importance is provided based on the index, if an old neuron has to be removed all newer neurons must be pruned as well, this limits the neural network from moving away from features it learnt.
2. More redundant copies, the importance acts as a weight to the activation, therefore when the strength of an important neuron might not be enough, the neural network may choose to learn redundant copies to strengthen the importance of features. This may also lead to superposition of very important features on a small subset of the layer.
3. If there are two features A and B being computed at index x and x+1, if B becomes more important than A as the training progresses, then either both A and B are dropped and re-learnt (which is hard if the index is small) or A transforms to B and B transforms to A over multiple iterations, which can slow convergence. The main drawback is we cannot effectively rearrange importance of neurons because of the hard index based importance.
4. While appendix H shows that the learnt width gets similar performance when trained from scratch while fixed, is this the best width for the accuracy? could an ablation be done when training fixed neural networks with an increasing width in intervals of 100 until 1.5/2 x width.

**Questions:**

1. A neuron can be less important for a specific subset of data but important for another subset of data, limiting this spike by applying an expected importance of the neuron , can lead to multiple neurons computing the same features to circumvent the loss in spike, this phenomenon can increase the number of redundant parameters being learnt. If we randomly drop certain neurons with similar importance do we observe minimal drop in performance?
2. Why did the paper not explore Adaptive Depth, is it because of stability issues (like in case of depth, the deeper the layer is its expected to be more important)?

---

> ### Author Response · Authors · 2025-11-17
> **Response to Reviewer KoUG: Part 1**
>
> We sincerely appreciate the interest the Reviewer showed in our work with this in-depth review. The questions and comments touch the very core functioning of our technique, and we are happy to provide further clarifications, starting from the questions:
>
> **Question 1:** The behavior the Reviewer references is highly nontrivial to analyze qualitatively. We address the specific question below, but we also refer the Reviewer to our comments and analyses regarding Weakness 2 for additional insights. When we remove adjacent neurons with similarly high performance, we observe that performance sometimes does not decrease, which does suggest potential redundancy. However, we cannot conclude at this stage that this behavior is directly attributable to the Reviewer’s conjecture.
>
> **Question 2:** The combination of AWNN with the adaptive depth of Nazaret and Blei (2022) is an extremely interesting future direction we already had in mind, but we did not do it yet for two reasons: i) we want to prioritize our understanding of how AWNN works before moving on, which is why our work contains different qualitative experiments; ii) the combination will have impacted the presentation of the novel methodology, which is the adaptive-width. That said, we do not expect particular obstacles since both techniques are generally stable.
>
> ----
>
> Now please allow us to comment on the weaknesses highlighted by the Reviewer.
>
>
> **Weaknesses 1 and 3:** Though we did empirically observe comparable convergence to standard networks on various tasks (Figure 2 right, Figure 9), we understand what the Reviewer means. We can offer a consideration: what the Reviewer saw as a “disadvantage” we perceived it as a feature, in the sense that enforcing an ordering of neurons is also a way of reducing equivalent parametrizations up to a permutation of the neurons, thus avoiding the jostling effect (line 65). Please also refer to point 1 of Reviewer DdyY where we have investigated other distributions, such as a soft step function. For this type of distribution, we expect these effects to diminish as nearly all neurons are weighted uniformly up to the transition point. Generally, we agree that stable convergence may not mean optimal convergence and this thus requires further investigation. However, we note that this is also a common problem of classical fixed-width neural networks. We did not study these aspects in great detail in this paper, especially given space limitations, but we see it as an interesting direction for the future, especially in the context of network alignment.
>
> **Weakness 2:** This is related to question 1. It could happen in principle (except for cases where we both regularize weights and bound activations, as discussed in Section 3) that extra copies are created the way the Reviewer suggests. Yet, because there is a finite budget of important neurons, one expects that the neural network will not prioritize copying the same “hidden feature” just to increase the activation of it.
>
> In an effort to shed more light into this behavior, we ran two analyses in addition to Figure 5 (left), using the same neural network. The first analysis checks what happens if we truncate the network in the opposite order (from most to least importance) with respect to Figure 5: performance quickly drops as expected, and random performance is reached after dropping the first ~40 neurons out of 83. The second analysis prunes individual neurons to understand how much their absence influences performance (which can have a different effect compared to removal of a set of neurons), to see if there exist two neighboring neurons where the removal of one contributes minimally to the performance drop as envisioned by the Reviewer. As expected, the first 25 neurons out of 83 are the main responsible for the good performances, but interestingly the removal of some neurons does not affect accuracy. Instead, removing subsequent neurons has little impact on performances, as one would expect from a good importance ordering.
>
> **These analyses are now reported in a new appendix P in the revised paper.** They provide some insights about the correlation between neurons and accuracy, but in the end we think it is still hard to fully verify the conjecture of the Reviewer. This is because it is not necessary that copies are adjacent to amplify a specific feature, and because it is unclear to us how to define different “subset” of data with similar/different features. In this sense, we feel the complexity of identifying these patterns in AWNN networks would be equivalent to classical ones.
>
>
> Of course, the investigations required to fully understand this new methodology and its implications cannot be compressed into a single conference paper. Our work focuses mainly on showing that this novel general methodology can work in the first place.

---

> ### Author Response · Authors · 2025-11-17
> **Response to Reviewer KoUG: Part 2**
>
> **Weakness 4:** The objective of Appendix H was twofold: investigating whether the width learned by AWNN displayed better standard deviation or accuracy when plugged into a fixed model, and showing how human bias can lead to underperforming models by selecting the wrong range of widths to try for a task, as happened in the REDDIT-B case. In fact, an approximation for the best width of fixed models was already shown in Table 1 by performing hyperparameter-tuning with the ranges typically selected by researchers before us. For these reasons we do not think there is a weak point here. Please let us know if we misunderstood your request.
>
> However, since the best approximation for the width in the case of REDDIT-B is unknown to us, as we copied results from Errica et al. (2020), we ran the experiment as the Reviewer suggested. In particular, we took the best configuration chosen during model selection by AWNN, and then tried to train a fixed model with different widths up to $\sim 2\times$ the chosen width (an average width of 793 neurons spread across 5 layers, meaning 158 neurons per layer on average in the fixed network). We report the results below:
> | Total Width | Mean  | Std  |
> |------------------|-------|------|
> | 50               | 89.28 | 3.50 |
> | 250               | 92.89 | 1.45 |
> | 500              | 92.67 | 2.03 |
> | 793              | 92.83 | 3.55 |
> | 1000              | 93.06 | 1.60 |
> | 1250              | 92.78 | 2.33 |
> | 1500              | 91.78 | 2.41 |
>
> Though these results are statistically comparable, the average performance obtained using AWNN’s learned width seems to be close to a local optimum for the fixed-width model. We have included the results in the Appendix to strengthen the paper. Thank you for suggesting this analysis!

---

> ### Comment · Reviewer_KoUG · 2025-11-26
> **Thanks for the rebuttal**
>
> Thanks for the rebuttal, I stick with my original score

---

### Official Review · Reviewer_DdyY · 2025-11-03

**Soundness:** 3
**Presentation:** 4
**Contribution:** 3
**Rating:** 6
**Confidence:** 4

**Summary:**

This paper proposed a Bayesian method for setting the number of neurons per layer in a neural network. It starts with a (theoretically) infinite-width neural network and trains both the weights/biases and a width parameter for each layer. The chosen Bayesian method is Variational Inference.

Summary of a **theoretical** training step:
1. sample the width parameters $\lambda_l \sim \mathcal{N}(\nu_l, 1)$;
2. use the $\lambda_l$ to compute the neuron scalings $(f_l(i))_{i \in \mathbb{N}^+}$, where $f_l$ is a decreasing function parameterized by $\lambda_l$ (the larger the $\lambda_l$, the quicker $f_l$ decreases);
3. given the $(f_l)_l$, compute the practical widths $(D_l)_l$: $D_l$ is the smallest integer such that $\sum_{i=1}^{D_l} f_l(i) \geq k = 0.9$;
5. depending on the new widths $D_l$, prune or create neurons at each layer;
6. provided that each neuron outputs $h_j^l = f_l(j) \, \sigma(\sum_{i = 1}^{D_{l-1}} w_{ji}^l h_i^{l-1})$, that is, the output of the activation function $\sigma$ scaled by a factor $f_l(j)$, perform a forward-backward and gradient step on the variational parameters $\rho_{ji}^l$ of the weights $w_{ji}^l \sim \mathcal{N}(\rho_{ji}^l, 1)$.
In practice, the variational parameters $(\nu_l)_l, (\rho_{ji}^l)_{lij}$ are trained directly, **without sampling** of the width parameters $\lambda_l$ or the weights $w_{ji}^l$.

The neuron scalings $f_l(i)$ are a key component of the method, since they allow:
* to practically deal with theoretical infinite-width layers;
* to backpropagate the gradient of the loss up to the variational parameters $\nu_l$: the loss does not depend continuously on the $D_l$, so $D_l$ cannot be used directly to optimize the loss. But $(f_l(i))_{li}$ works as a proxy for the "layer width" and: a) we have a continuous dependence of the loss on the $f_l(i)$; b) the $f_l(i)$ can be randomly generated with a continuous dependence on the variational parameters $\nu_l$.

**Strengths:**

# Clarity
The paper is well-written.

# Significance
This paper proposes a Bayesian formulation of the problem of width selection when training a neural network, which is a well-grounded way of performing pruning and adding neurons (layer-wise). Additionally, the fact that, in theory, infinite-width neural networks are trainable (up to a decreasing scaling factor $f_l(i)$), makes this method fit several well-known theoretical frameworks (e.g., Neural Tangent Kernels).

# Novelty
There are several Bayesian methods to select neurons or parameters, but, up to my knowledge, the method as a whole is new.

Please note that the idea of considering possibly infinite neural networks with decreasing scaling factors is not new. See for instance [1], which also cites [2] as a work introducing "the idea of using asymmetrical scaling parameters", which date respectively from 2023 (published in 2025) and 2020.

However, the proposed method is a unified and original way to remove and add neurons, which is undeniably new. Additionally, the idea of **parameterized** asymmetrical scalings is new.

[1] *Over-parameterised Shallow Neural Networks with Asymmetrical Node Scaling: Global Convergence Guarantees and Feature Learning*, F. Caron et al., Transactions on Machine Learning Research, 2025.

[2] *Asymmetrical Scaling Layers for Stable Network Pruning*, P. Wolinski et al., [https://openreview.net/pdf?id=6GUIv9eYnD7](https://openreview.net/pdf?id=6GUIv9eYnD7), 2020.

**Weaknesses:**

I do not see major weaknesses in this paper. But, on the technical side, several aspects deserve a discussion :
1. why choosing a discretized exponential distribution for $f_l$? Is there a theoretical/heuristic argument? One could choose, as in [2], slowly decreasing scalings, such as $x \mapsto 1/x$ or $x \mapsto 1/(\sqrt{x} \ln(x))$...
2. according to Eqn. (7), $\lambda_l$ could be either positive or negative, since it is a Gaussian random variable. This is not acceptable, provided Eqn. (6), where $\lambda_l$ is a scaling factor in an exponential. Is this a problem in practice? Even if it is not, would it be possible to replace the Gaussian distribution by a distribution supported on $\mathbb{R}^+$?
3. p 4, line 179: "Appendix A and B provide formal requirements about $f_l$". It seems that these Appendices are not about such "formal requirements", they are about "Background notions" on variational inference (App. A) and "How to compute $D_l$ in practice" (App. B). I am very curious about these "formal requirements", would it be possible to provide some? Anyway, this reference to Appendices A and B is not consistent with their content.

Clarity: are some "minus" missing in the exponentials in Eqn. (6)? It would be nicer to write $e^{-\lambda_l x}$ instead of $e^{\lambda_l(x)}$ (removing the parentheses would also makes it easier to read).

Style (minor concern): In the abstract and in the introduction, the authors **seem** to claim that there is no research in automatic hyperparameter search (which includes hyperparameters related to neural network architecture). The authors write "For almost 70 years, researchers have **primarily** relied on hyper-parameter tuning [...] This paper challenges the status quo [...]", which is technically correct, but slightly misleading. It is more a matter of style, which may be seen as a little fancy, than a matter of knowledge of works on neural architecture search, which are cited.

**Questions:**

See weaknesses.

---

> ### Author Response · Authors · 2025-11-17
> **Response to Reviewer DdyY**
>
> We would like to thank the Reviewer for the constructive feedback, the on-point questions, and for raising a few points that require clarification. Also, thanks for the additional references, which we have included in the related work section. We hope to engage in a meaningful discussion and answer to the points raised one by one below.
>
> **Point 1:** The choice of the exponential distribution was driven by: (i) its simplicity; (ii) the natural decay in importance across subsequent neurons, which we considered a necessary inductive bias in this context; and (iii) the ability to compute its quantile function in closed form. In this sense, we agree that future investigations into the behavior of importance functions are necessary to understand how this choice influences convergence (for instance, we initially experimented with a power-law distribution, but convergence was noticeably slower). In this work, our primary focus was on demonstrating that the proposed methodology can work in the first place.
>
> To address the Reviewer’s curiosity, we conducted another experiment on SpiralHard where, in addition to the exponential distribution, we tested two additional distributions with different characteristics:
>
> - A Power Law distribution with starting gamma values in [2., 3.] and low degree saturation chosen between [0., 2.]. Here we learn the gamma value.
> - A sigmoidal function modelling a soft step function with a starting transition point in [128, 256]. Here we learn the transition point.
>
> The results are reported below. Across all hyper-parameter configurations (Table 4 in the revised PDF), the Power Law performs slightly better than the Exponential but introduces substantially more neurons due to its long tail. In contrast, the Sigmoidal function allocates fewer neurons and, on average, yields slightly lower validation performance. One drawback of the latter is the loss of the ability to truncate the network after training, since the importance values remain very close to 1 before the transition point, and the transition itself, given the chosen fixed parameters, is relatively sharp. We note that the standard deviations are higher than usual because certain hyper-parameter configurations are simply suboptimal relative to others.
>
> | Base Distribution               | Mean Score | Std Score | Min Score | Max Score | Mean Total Width | Std Total Width |
> |--------------------------------|------------|-----------|-----------|-----------|-------------------|------------------|
> | Exponential        | 80.27      | 19.90     | 57.25     | 100.00    | 954.39            | 1083.90          |
> |Power Law           | 81.82      | 16.59     | 58.63     | 100.00    | 2952.44           | 3371.60          |
> |Sigmoid | 76.85      | 18.29     | 57.63     | 100.00    | 426.78            | 268.41           |
>
> We have added these results to the main text of the revised paper using the extra page available. Thank you for suggesting this analysis, which helped improve the paper by showcasing how different importance functions have different properties.
>
>
> **Point 2:** Thanks for catching this inconsistency. The formulation was indeed too general for the specific choice of the exponential. In practice, this is not an issue as long as the values of $\nu$ are guaranteed to be positive, which we enforce through a softplus activation. However, from a more formal standpoint, one may replace the Normal distribution with a Folded Normal distribution, which has support only on the positive real numbers. We have updated the paper to make this clearer.
>
> **Point 3:** We agree that the use of the term “formal requirements” was ambiguous and apologize for the confusion. What we meant is that in Appendix A, Def A.2, we tried to define desirable properties for the distribution $f(\lambda)$ – which we called $q(\omega)$ for consistency with Def A.1 but should be changed to $f(\lambda)$. We also agree that Appendix B should not be cited in that line. We have revised the paper accordingly. We hope that our answer clarifies the doubt, but please let us know if this is not the case.
>
>
> **Clarity:** thank you for spotting the typo! We have modified the paper as suggested.
>
>
> **Style:** We agree that, despite a stylistic choice, this deserves clarification. We have adjusted the abstract as follows: *“...have typically selected the width of neural networks’ layers either manually or through automated hyperparameter tuning methods such as grid search and, more recently, neural architecture search”*. This way we mention more explicitly previous efforts in the automatic hyper-parameter tuning space, but we also imply that the width is not considered a hyper-parameter in the present work.
>
> -----
>
> **Conclusions:** We hope that our clarifications addressed the Reviewer’s doubts, and we would sincerely appreciate it if the Reviewer would consider increasing the score as a result. The suggestions were on point and further strengthened the paper, thank you very much!

---

### Author Response · Authors · 2025-11-17
**General Response to Reviewers**

We want to sincerely thank all Reviewers for their constructive feedback which helped us strengthen the paper.

**We have uploaded a revised version of the manuscript following your suggestions. The blue text corresponds to the changes made.** Instead, please refer to our individual answers to you for the details.

We would greatly appreciate it if the Reviewers could consider raising their scores in light of our efforts.


Best regards,

The authors

---

### Comment · Area_Chair_46t4 · 2025-11-24

Dear reviewers,

       The authors now have given their response to the reviews, please have a look on the rebuttal and revised PDF.

      After that, please give your final rating on this submission.

Your AC
Best

---

### Author Response · Authors · 2025-12-01
**Message to (New) AC**

Dear (new) AC,

For the reasons below, we thought it could be a good idea to share with you some information about the progress of our rebuttal, though we understand that the AC has no obligation towards what happened prior to the reassignment of papers to ACs.

We are aware that the recent leak may have affected, in principle, our submission. However, we think there is no indication that our Reviewers artificially raised their scores, since the reviews we received were **detailed, professional**, and the Reviewers' confidence seemed adequate to their evaluations.

 For these reasons, we think it is important to share with you that, **prior to the PCs action:**

- Reviewer DdyY did increase the score to 8 and raised confidence to 5
- Reviewer SBvZ raised the score to 6

This happened **on Nov 24th, shortly after the previous AC asked the Reviewers to take into account our rebuttal responses**. Therefore we believe there was a clear relation between the ACs message and the Reviewers’ update.
Since the leak was made public on Nov 27th, clearly after the Reviewers actions, we do not believe our Reviewers were affected by the leak. **This message is intentionally visible to all involved parties.** In any case, we kept screenshots but are aware that these could be fabricated and cannot be uploaded here.

We would appreciate it if you could take our considerations into account for the final decision.

Best regards,
the authors

---

### Meta-Review · Area_Chair_3eNs · 2025-12-19

**Summary:**

Reviewers unanimously acknowledged that the paper tackles a very important research question, adaptive width in neural networks, and that the proposed method is sound and meaningful. They agree that the simultaneous learning of the NN's width allows for an efficient practical approach. The two main points of criticism are a close relation to some prior works and a limited empirical evaluation in terms of comparison with alternatives. It is further noteworthy that one reviewer indicated low confidence in their assessment and communicated this very clearly to the AC, in particular regarding the technical aspects. The AC has carefully considered the paper, the rebuttal, and the discussion that had already happened prior to the freeze this year, weighing in the different levels of confidence.
Overall, the AC believes that the paper provides an important and meaningful contribution, where the benefits presently outweigh the limitations in terms of limited empirical comparison. As such, the AC recommends to accept the paper as a poster.

**Reviewer Concerns:**

Each reviewer had different concerns, ranging from requests for thorough empirical comparisons, to more detailed methodological explanations, and lastly discussion related to other works. The reviewer who gave the lowest score out of the four was simultaneously the reviewer with the lowest confidence. A respective rebuttal has provided clarifications, which the AC believes would have resolved some of the questions, even if the concern on empirical comparisons to other methods would likely still hold.
For the three other reviewers, the rebuttal appears to resolve the most critical questions and remarks, with the exception of reviewer A2U1, whose very short review leaves little room for discussion. Overall the AC believes that some aspects may be outstanding, but that the vast amount of concerns has either been clarified or has explicitly been indicated to be of lesser importance by the reviewer themselves (e.g. giving a score of 8 ,raising concerns, but deciding to stick to the rating)

**Reviewer Scores:**

Initial reviewer scores were 6,8,4,6, whereas the score of 4 is associated with a very low confidence rating. Two reviewers were able to participate in the discussion, resulting in one of the reviews with a 6 raising the score to 8 prior to the rollback (through an edit of the original review with comment that is only visible in the revision history). The reviewer who gave an 8 also commented that they are satisfied with the response. The AC believes that the review with the score 4 would have potentially raised their score to 6 following the clarifications in the rebuttal, or alternatively, nevertheless recommended to accept the paper according to the higher indicated expertise of the remaining reviewers. The final reviewer who gave a score of 6 gave an exceptionally short review. It is unclear whether any discussion would have occurred here and whether it would have resulted in any kind of change in scores. However, even without this fourth review, the AC believes that the picture for this paper is clear and recommends acceptance.

---

### Decision · Program_Chairs · 2026-01-26

Accept (Poster)